# Polymersomes as the Next Attractive Generation of Drug Delivery Systems: Definition, Synthesis and Applications

**DOI:** 10.3390/ma17020319

**Published:** 2024-01-08

**Authors:** Mariana Fonseca, Ivana Jarak, Francis Victor, Cátia Domingues, Francisco Veiga, Ana Figueiras

**Affiliations:** 1Univ. Coimbra, Laboratory of Drug Development and Technologies, Faculty of Pharmacy, 3000-548 Coimbra, Portugal; mariana.fons@hotmail.com (M.F.); jarak.ivana@gmail.com (I.J.); cdomingues@ff.uc.pt (C.D.); fveiga@ff.uc.pt (F.V.); 2Instituto de Investigação e Inovação em Saúde, University of Porto, 4200-135 Porto, Portugal; 3Department of Pharmacy, University Chenab Gujarat, Punjab 50700, Pakistan; vfrancis030@gmail.com; 4Univ. Coimbra, REQUIMTE/LAQV, Group of Pharmaceutical Technology, 3000-548 Coimbra, Portugal; 5Univ. Coimbra, Institute for Clinical and Biomedical Research (iCBR), Area of Environment Genetics and Oncobiology (CIMAGO), Faculty of Medicine, 3000-548 Coimbra, Portugal

**Keywords:** polymersomes, drug delivery systems, preparation methods, cancer therapy, chemotherapy, immunotherapy, self-assembly process, block copolymers, regulatory issues

## Abstract

Polymersomes are artificial nanoparticles formed by the self-assembly process of amphiphilic block copolymers composed of hydrophobic and hydrophilic blocks. They can encapsulate hydrophilic molecules in the aqueous core and hydrophobic molecules within the membrane. The composition of block copolymers can be tuned, enabling control of characteristics and properties of formed polymersomes and, thus, their application in areas such as drug delivery, diagnostics, or bioimaging. The preparation methods of polymersomes can also impact their characteristics and the preservation of the encapsulated drugs. Many methods have been described, including direct hydration, thin film hydration, electroporation, the pH-switch method, solvent shift method, single and double emulsion method, flash nanoprecipitation, and microfluidic synthesis. Considering polymersome structure and composition, there are several types of polymersomes including theranostic polymersomes, polymersomes decorated with targeting ligands for selective delivery, stimuli-responsive polymersomes, or porous polymersomes with multiple promising applications. Due to the shortcomings related to the stability, efficacy, and safety of some therapeutics in the human body, polymersomes as drug delivery systems have been good candidates to improve the quality of therapies against a wide range of diseases, including cancer. Chemotherapy and immunotherapy can be improved by using polymersomes to deliver the drugs, protecting and directing them to the exact site of action. Moreover, this approach is also promising for targeted delivery of biologics since they represent a class of drugs with poor stability and high susceptibility to in vivo clearance. However, the lack of a well-defined regulatory plan for polymersome formulations has hampered their follow-up to clinical trials and subsequent market entry.

## 1. Introduction

According to the European Medicines Agency (EMA), nanotechnology involves “the use of structures less than 1000 nanometers across, which are designed to have specific properties”. Among many types of nanodelivery systems of active pharmaceutical ingredients (API), liposomes are already used in clinics and have been described in great detail in recent decades [1]. They are biocompatible and show low in vivo toxicity, and their use is based on the improvement of treatment efficiency. However, liposomes are associated with drawbacks such as limited chemical, physical, and biological stability, which hamper their use in drug delivery [2]. In this regard, polymeric vesicles or polymersomes present a promising alternative to overcome these disadvantages [3]. Both polymersomes and liposomes are composed of a hydrophilic core and a hydrophobic bilayer, being able to incorporate hydrophilic and hydrophobic drugs. However, polymersomes have tunable thickness and membrane fluidity, allowing controlled release. They also possess better stability in the biological environment [4], allowing for improved pharmacokinetic and pharmacodynamic profiles of APIs and, consequently, their enhanced therapeutic performance [5]. Moreover, they can be tuned to provide better oral bioavailability, improved absorption, and increased drug release relative to liposomes [4]. Despite the advantages, polymersomes also have potential weaknesses, such as low biocompatibility [6] or permeability [7]. However, the versatility of copolymers makes it possible to overcome these problems, and polymersomes with distinct functional characteristics have been designed, such as fluorescent-labeled [8,9], ligand-decorated [10,11], or stimuli-responsive polymersomes [5,12,13]. The versatility of building blocks combined with processing parameters can also give rise to a variety of structural features, and tubular (tubesomes) [14], porous [15], and giant polymersomes [6] have been prepared along with the classical spherical bilayer nanosized vesicles [16].

Polymersomes can be assembled from a great variety of copolymers of different structural and physicochemical characteristics. Although certain structural parameters are required for successful polymersome formation, this variety provides endless possibilities for combining functional elements that enable size and shape control, targeted delivery, and stimuli-sensitive drug release. With the variety of building blocks, various fabrication techniques have been developed that can complement the control of polymersome characteristics by fine-tuning producing parameters and broadening the choice of loaded active therapeutic ingredients. The compartmentalized structure of polymersomes offers possibilities for the encapsulation of both hydrophobic (within the hydrophobic layer) and hydrophilic (within the aqueous core or on the shell surface) molecules. Modalities that enable the stabilization of polymersomes by cross-linking or are sensitive to biological or external (physical) stimuli enable control over the drug release rate or spatiotemporal release.

There has been increased investment in this field, and its potential has improved during the past few years. Among many medicinal applications, antibacterial therapy [17,18], vaccine development [19,20], diabetes mellitus [21], trimethylaminuria (TMAU) (“fish odor syndrome”) [22], and cancer therapy [23,24,25] are only some areas that could potentially benefit from these nanocarriers. Polymersomes as drug delivery systems can be good options to overcome therapeutic problems because they allow encapsulation of therapeutic agents, protecting them from the biological system and driving them to the exact site of action, preventing them from reaching off-target parts of the body and reducing toxicity issues at the same time. Moreover, it is also possible to encapsulate more than one API in the same system, allowing the delivery of several therapeutics simultaneously to improve efficacy, decrease side effects, and avoid drug resistance to therapy [26]. Nevertheless, the delivery of biotherapeutics shows limitations relative to structural complexity, stability, immune system activation, and cellular uptake that justify their encapsulation within nanocarriers [27], and polymersomes have also been successfully applied in the delivery of this type of therapeutics.

The design and application of polymersomes is a dynamic multidisciplinary field, spanning material science, technology development, and biological application. Therefore, this work presents the advances made in the delivery of therapeutic molecules in this decade. Furthermore, some of the latest preparation methods that have been used and developed in recent years, as well as the range of block copolymers with distinct characteristics that have been designed for polymersome preparation will be addressed. Moreover, the most recent examples of their application relative to biotherapeutic delivery will also be summarized. Finally, aspects related to nanosystem regulatory issues will also be delineated and discussed.

## 2. Polymersome Definition and General Considerations

Polymersomes, also called polymeric vesicles, are artificial synthetic structures composed of amphiphilic block copolymers (Figure 1a) [28]. Constituting hydrophobic and hydrophilic blocks of desired characteristics [29] enable the incorporation of hydrophilic molecules in their aqueous core and/or hydrophobic compounds in the interior of the polymersome membrane, as demonstrated in Figure 1b [8]. The interfacial tension between hydrophobic chains and water molecules allows the self-assembly of copolymers in aqueous medium into kinetically stable vesicles [4]. Over the last few years, their characteristics have been improved, with great potential in areas such as drug delivery and diagnostics, or as nanoreactors used for chemical synthesis or sensing [30]. The formation of polymersomes can be influenced by molecular curvature and the packing parameter dictated by the composition of amphiphilic block copolymers (the ratio between the optimal polymer area resulting from the balance between the hydrophobic and hydrophilic interactions, and hydrophobic chain volume and chain persistence length), and also by the kinetics of the polymersome formation process [31]. The morphological characteristics of nanoaggregates can influence cellular uptake, drug delivery, and the effect of cytotoxic drugs in cells [32]. Most of the studies included in this review paper report the formation of polymersomes as a consequence of copolymer structure and production conditions. Although great control over aggregate morphology can be achieved by structure/process optimization, under certain conditions a complex mixture of products can be obtained. When poly(2-(methacryloyloxy)ethylphosphorylcholine) was used as a hydrophilic component in a PMPC_25_–PDPA_70_ copolymer, both spherical and tubular polymersomes were obtained alongside spherical micelles [33]. The authors demonstrated that various separation techniques could isolate different shapes and structures. The influence of purified spherical and tubular polymersomes on physiological processes such as internalization mechanisms and molecular pathways demonstrated cell-specific response to each shape. Doxorubicin (Dox)-loaded tubular polymersomes showed higher toxicity against cancer cells compared to spherical polymersomes [32]. Similarly, the elongated shape of prolate polymersomes promoted cellular uptake compared to spheric polymersomes [34]. 

Another aspect to consider is the nanoparticle size. While the choice of morphological parameters may control the morphology and wall thickness of the polymersomes, the size of vesicles can be influenced by the formation process [28]. The fact that endothelial cells of tumor blood vessels have larger intercellular spaces facilitates the extravasation of nanoparticles from the bloodstream into the tumor site. At the same time, the absence of lymphatic vessels in the tumor tissue allows for the accumulation of polymersomes in the tumor microenvironment. These tumor features, known as the enhanced permeation and retention (EPR) effect, have long been considered the main mechanism of tumor retention [35]. In the last decade, they have been supplemented with additional mechanisms based on transcytosis and vascular burst [36]. Nanoparticles with diameters exceeding 200 nm tend to accumulate in the liver and spleen, while those smaller than 200 nm can more easily accumulate in the tumor due to the EPR effect following intravenous administration since they are not so easily detected by the reticulum endoplasmic system (RES) [24,26]. 

The stability of nanoaggregates is indispensable for prolonged systemic circulation and can prevent the uncontrolled loss of APIs. Apart from controlling the stability by the choice of copolymer composition, it can be enhanced by membrane crosslinking. For example, polystyrene PEG-PS vesicles have the disadvantage of disassembling under harsh conditions or when polar solvents are used [37]. To prevent disassembly, the introduction of acrylate groups in poly(ethylene glycol)-*b*-poly(styrene-*co*-4-vinylbenzyl acrylate (PEG-*b*-P(S-*co*-4-VBA))) enabled UV-induced crosslinking and structure stabilization [37]. Spontaneous crosslinking of sulfur-containing moieties is also often exploited for prolonged storage and dilution stability [38]. 

Equally important is the ability of polymersomes to release their therapeutic cargo in a time- and space-dependent manner. Drug release mostly occurs via passive diffusion, and the bilayer membrane’s porosity for the controlled release can be adjusted by its thickness, composition, or crosslinking [39]. The introduction of phase separation within the polymersome membrane is attracting increased attention since it leads to the formation of inhomogeneous membrane porosity [40]. In a study by Chen et al., the phase separation was accomplished by combining semicrystalline polycaprolactone (PCL) and amorphous poly((α-(cinnamoyloxymethyl)-1,2,3-triazol)caprolactone) (PCTCL) units, and was assisted by crosslinking to achieve tunable release rates of Dox [40]. Other approaches can also be used to manipulate membrane porosity. The formation of pores in the membranes of polymersomes to control the porosity can be achieved via the incorporation of pore generators (porogens). An example of this is the co-assembly of an inert block copolymer PEG-*b*-PS and a stimuli-responsive block copolymer poly(ethylene glycol)-*b*-poly-(acrylbenzylborate) (PEG-*b*-PABB) [15]. The latter, when added in a lesser quantity and when exposed to stimuli (hydrogen peroxide), undergoes dissolution of a stimuli-responsive moiety, while the inert PEG-*b*-PS remains in the structure leading to pore generation [15]. An alternative strategy is the controlled disassembly of polymersomes via the application of stimuli-sensitive polymers, which will be discussed in detail in the following sections. 

Besides the parameters listed above, the mechanical properties of polymersomes can also determine the biological applicability of polymersomes. In a library of polymersomes of identical size, shape, and zeta potential, less elastic particles possess better blood–brain barrier (BBB) and tumor penetrability as well as improved tumor cell uptake than their more elastic counterparts [41]. 

Surface modifications of polymersomes can have a great impact on their surface characteristics and functions, as will be demonstrated in subsequent sections in greater detail [35]. 

Polymersomes show a great variability of properties, presenting advantages and disadvantages alike. The latter can be optimized/minimized by improving their structure and properties. Some advantages and disadvantages associated with polymersomes, as well as examples of the strategies that can be used to tackle the observed shortcomings, are summarized in Table 1. 

## 3. Different Types of Polymersomes

Polymersomes can be endowed with high functional versatility relevant to diagnostic and therapeutic applications owing to their synthetic plasticity, and three large groups may be highlighted. One contains polymersomes labeled with imaging molecules. The second includes polymersomes decorated with targeting ligands for selective delivery. The last class of functionalized polymersomes is based on stimuli-responsive polymersomes, able to transport and release therapeutic molecules in a controlled manner and in response to external (light or temperature) or internal (pH, redox potential, or enzymes) stimuli. 

### 3.1. Copolymer Types

There are many copolymers with different compositions, characteristics, and pathways of synthesis, with a characteristic composition that allows the acquisition of polymersome morphology with desired properties to be used in drug delivery systems and other applications. Some of them are described below and are shown in Table A1 (Appendix A). 

The most used hydrophilic copolymer is biocompatible poly(ethylene oxide) (PEO), also known as poly(ethylene glycol) (PEG) [43]. Adjustable length, density, and configuration of the PEG layer can protect a therapeutic load under physiological conditions and delay clearance from the bloodstream [28]. However, it has some drawbacks, including the use of toxic and gaseous monomer ethylene oxide in polymer synthesis, risk of peroxidation, possible immune response, and protein adsorption [44]. A recent investigation of the influence of terminal PEG OH groups on the formation of protein corona and polymersome uptake by diverse immune cell subpopulations in different biological compartments could provide valuable insights for the rational design of polymersomes, especially in the field of immunotherapy [45]. Alternative hydrophilic polymers have also been explored, such as poly(oxazoline)s, poly(sarcosine), and poly(glycidol), or oligosaccharides [46], because of their biocompatibility and structural and synthetic versatility [47]. At the same time, they show enhanced hydration and better antifouling properties [43]. When PDPA was conjugated with (poly([*N*-(2-hydroxypropyl)]metha acrylamide)) (PHPMA) hydrophilic moiety, the resulting PHPMA_35_-*b*-PDPA_42_ block copolymer offered advantages in terms of its protein-repelling properties [31]. Relative to PEG-*b*-PLA, PHPMA_35_-*b*-PDPA_42_ block copolymer showed higher stability with inconsiderable protein binding, while PEG-*b*-PLA polymersomes proved to be susceptible to protein adsorption in a protein-dependent way. In this case, the length of the hydrophilic shell and the chemical nature of the outer surface had the highest influence on the protein adsorption [31]. Although the influence of PEGylation parameters has been well studied in various nanosystems, comprehensive studies of the same kind are rare for other types of hydrophilic polymers. To fill this gap, Najer et al. evaluated the influence of poly(2-methyl-2-oxazoline) length on the antifouling properties of the poly(2-methyl-2-oxazoline)-block-poly(dimethylsiloxane)-block-poly(2-methyl-2-oxazoline) (PMOXA-*b*-PDMS-*b*-PMOXA)-based polymersomes [1]. Mixtures of polymers with hydrophilic fractions between 43 (n_21-65-21_) and 20 (n_6-65-6_) % (the limit values for polymersome formation based on f_hydrophilic_ of ≈35 ± 10%) were used. Maximizing PMOXA length and amount within the mixture allows for maximization of interfacial water layer thickness and density, optimizing the antifouling behavior. Nonetheless, PMOXA length remains bound to the length of the hydrophobic component in the copolymer in order to ensure the formation of stable polymersomes. Although the formulation with the least protein corona performed the best in in vitro and in vivo models, it was inferior to PEGylated liposomes of similar size, offering possibilities for further optimization of polyoxazolines. 

The hydrophobic components of polymersomes have also been studied, and special attention has been paid to biodegradable polymers with low in vivo toxicity. They include poly(lactic acid) (PLA), PCL, and poly(trimethylene carbonate) (PTMC) [28]. A comparison of poly(propylene oxide) (PPO) and poly(butylene oxide) (PBO) concluded that PBO has improved cytocompatibility, higher hydrophobicity, simpler fabrication, and lower glass transition temperature (−70 °C), which is useful when high membrane fluidity and flexibility are needed [43]. It makes PBO advantageous in relation to other hydrophobic blocks, such as PPO, PLA, or PCL. PLA and PCL need high temperatures during the self-assembly process since they are semi-crystalline and, therefore, cannot be used with biologically active compounds sensitive to high temperatures. Other hydrophobic moieties can also be used. Cholesterol (Chol) and α-tocopherol are biological, lipophilic, and biocompatible molecules used for the core construction of self-assemblies [48,49]. Polystyrene (PS) is a hydrophobic, FDA-approved copolymer that can be used for API-free anticancer therapy and is another example of a hydrophobic block often used for polymersome construction [50,51,52].

The addition of charged polymers can influence the interactions of polymers with both therapeutic loads and biological compartments like cell membranes. They are widely used for the complexation of therapeutic nucleic acids and some examples will be explored in Section 6. However, some positively charged polymers can have biological activity in their own right, as demonstrated in the example of PCL-poly(lysine-stat-(S-aroylthiooxime) polymersomes [53]. PolyLys is a positively charged antimicrobial peptide [54] that promotes bacterial death and cellular uptake via interactions with cell membranes. Incorporation within a copolymer provides increased peptide stability and at the same time serves as a hydrophilic block. The s-aroylthiooxime (SATO) component serves as a source of H_2_S, which promotes cell migration and adhesion and has great potential in diabetic wound healing [55]. Since SATO is hydrophobic, incorporation into a copolymer increases its solubility and provides good dispersibility. When administered as a wound dressing spray (layer thickness 375 nm), the vesicular morphology was preserved and the addition of cysteine led to gradual H_2_S release. The combined effects of wound healing and bacterial inhibition observed in a murine model offer the potential for an efficient and simple platform for diabetic wound healing with good patient adherence.

Therefore, the combination of hydrophilic blocks with hydrophobic polymers and other functional structural elements can lead to the formation of novel polymersomes for API delivery.

### 3.2. Theranostic Polymersomes

One of the great conveniences of these nanosystems is the ability to introduce imaging/diagnostic tags that can be used in diseased cell/tissue imaging and/or to monitor their transport and distribution in the organism. In addition, they can also be used as therapeutic agents, like in photodynamic therapy, or can be combined with other types of APIs delivered by the same polymersome. 

One example is the fluorescence introduced by aggregation-induced emission (AIE) [56]. The low fluorochrome loading and fluorescence quenching upon aggregation are frequent phenomena associated with conventional fluorochromes. On the contrary, when luminogens are incorporated into the block copolymers, they aggregate and generate bright photoemission characterized by improved contrast and photo-stability, which enhance cell/tissue imaging [8]. An example of this type of imaging polymersomes is poly(ethyleneglycol)-block-poly(caprolactone-gradient-trimethylene carbonate) (PEG-P(CLgTMC)) loaded with the photosensitizer BODIPY [9]. Terminal blocks of tetraphenylethylene pyridinium allow preferential polymersome accumulation on the surface of mitochondria, a preponderant organelle for cell survival [8]. 

A recent study demonstrated that asymmetric cucurbit-shaped polymersomes could be used as nanomotors with a variety of potential applications, including drug delivery, cell/tissue imaging, or photodynamic therapy [9]. In this case, cucurbit polymersomes based on PEGylated AIEgenic poly(trimethylene carbonate) blocks PEG_44_-P(AIE)_5_ were coated with catalytic urease-based machinery via a layer-by-layer (LBL) technique (Figure 2a). When exposed to the fuel (urea), cucurbit polymersomes outperformed their spherical counterparts.

Chelator-free labeling with positron emission tomography (PET) radiotracers is preferable since it avoids chelator optimization and multistep modifications of the nanosystem. On the other hand, the incorporation of radiotracer within the hydrophilic cavity or hydrophobic layer of polymersomes can result in loss of tracer under physiological in vivo conditions. Coating of poly(*N*-vinylpyrrolidone)_5_-*b*-poly(dimethylsiloxane)_30_-*b*-poly(*N*-vinylpyrrolidone)_5_ (PVPON_5_-PDMS_30_-PVPON_5_) polymersomes with tannic acid provided a polyphenolic anchor layer for radiotracer ^89^Zr and antibody ligand trastuzumab for targeting HER-2-positive breast cancer [57]. Although the system was not applied for tumor imaging in vivo, adsorption of the polymersome surface demonstrated a marked influence on biodistribution (Figure 2b).

### 3.3. Polymersomes Decorated with Targeting Ligands

Polymersomes decorated with targeting ligands (Figure 1d) can improve their selectivity against specific targets, including immune and cancer cells, and could be used for intracellular API delivery, immunotherapy, or vaccines. An example of this approach is a PEGylated polymersome composed of PEG_22_-*block*-poly[(ε-caprolactone)_38_-*gradient*-(trimethylene carbonate)_37_] (PEG-p(CL-TMC)) decorated with immunoglobulin fragment Fc and CpG oligodeoxynucleotide, that target Fc receptors and activate TLR-9 receptors located on the endosomal membrane, respectively [20]. Surface PEGylation is a commonly used approach to improve the pharmacokinetic properties of nanosystems by decreasing interactions with immune system components such as the complement system and mononuclear phagocyte system [58]. However, conjugation with immune-cell-targeting ligands greatly increased interactions with and the uptake by immune cells. Both ligands demonstrated a marked increase in polymersome interactions with immune cells. Nonetheless, when the combination of both ligands was applied, the CpG ligand was the main driving force of cellular interactions, while the synergistic effects of both ligands on immune system activation were observed. This study demonstrates that PEGylated polysomes like PEG-p(CL-TMC) could be developed for potential use as vaccines. In another example of nanocarriers labeled with two targeting ligands, PEG-poly(D,L-lactic-co-glycolic acid) (PLGA) polymersomes loaded with curcumin and decorated with neuron-specific transferrin and Tet-1 ligands demonstrated neuroprotection and improved cognitive function in a murine model of Alzheimer’s disease [59].

To account for the limited amount of overexpressed receptors on the targeted cells and to prevent receptor saturation, one strategy is to include multiple ligands that target different receptors. This approach allows for the targeting of different cell populations in vivo [60]. On the other hand, to avoid targeting non-tumor cells expressing the same receptors, a single multivalent low-affinity ligand that target different receptors can be used [10]. This approach was used to improve the capacity of poly(2-(methacryloyloxy)ethyl phosphorylcholine)-poly(2-(diisopropylamino)ethyl methacrylate) (PMPC-PDPA) polymersomes to target the scavenger receptor class B member 1 (SRB1) and scavenger receptor class B member 3 (CD36) present on the surface of tuberculosis- and Staphylococcus aureus-infected macrophages or cancer cells [10]. This targeting capacity is based on the interactions with the phosphorylcholine groups (PC) of the PMPC chains, which promote the internalization of polymersomes by endocytosis. Taking advantage of these polymersomes to target monocytes, their use in cancer immunotherapy can be promising, taking into account that these cells represent 2−8% of the blood cells. 

A recent study by Tjandra et al. demonstrated the importance of ligand density optimization using the example of poly(ethylene glycol)-*block*-poly(*N*-isopropylacrylamide-co-perylene diester monoimide) (PEG_43_-*b*-P(NIPAM_21_-*co*-PDMI_9_)) ellipsoid polymersomes [61]. In this case, polymersome size (<100 nm or ~200 nm) did not influence cell uptake. However, a threshold density (four ligands per polymersome) of medulloblastoma cell (DAOY)-targeting ligand was required for enhanced uptake. Here, a highly specific targeting ligand, heptapeptide FSRPAFL, with high binding activity, was selected through a peptide phage display library. The non-linear tendency between ligand density and cell uptake was especially obvious in cells with many targeted receptors (DAOY compared to HEK) as the cumulative effect of interactions became much more pronounced. At the same time, the chosen heptapeptide provided stealth properties to the polymersomes, and reduced interactions with immune cells were observed when compared to control polymersomes. 

Another study described a composite methyl-poly(ethylene glycol)-polylactide (mPEG–PLA)/tocopherol-polyethylenelglycol-succinate (TPGS)-lactobionic acid (TLA) polymersomes equipped with TLA ligands that target asialoglycoprotein receptor (ASGPR) in hepatocellular carcinoma cells [11]. At the same time, this nanosystem hampers multidrug resistance by inhibiting P-glycoprotein, leading to improved internalization and intracellular accumulation of drugs. The iRGD peptide is also often used to decorate polymersomes and improve their binding to tumor vascular cells and selective internalization for targeted drug delivery [62,63].

The pandemic caused by SARS-CoV-2 elicited a rapid response and the development of vaccines. Although nucleic-acid-based nanovaccines have been approved for clinical use so far, subunit vaccines based on spike protein and its subdomains are also being developed. To improve humoral and cellular immunogenicity towards the receptor binding domain (RBD) of spike protein, Volpatti et al. prepared, as a new generation of vaccines, oxidation-responsive poly(ethylene glycol)-*bl*-poly(propylene sulfide) (PEG–PPS) polymersomes for delivery of antigen and adjuvant [64]. To evaluate the ability of PEG-PPS to activate B cells and induce the production of RBD antibodies, polymersomes were decorated with RBD (RBD_surf_) or encapsulated RBD in the polymersome cavity (RBD_encap_), and were supplemented with shell-encapsulated adjuvant monophosphoryl-lipid-A PS (MPLA PS). In vivo studies in the murine model demonstrated that only adjuvanted RBD_surf_ produced neutralizing IgG and RBD-specific germinal center B cells. Both surface and encapsulated RBD stimulated T cells (CD_4+_ and CD_8+_) and Th_1_-type cytokines. Furthermore, antigen/adjuvant-loaded systems are stable for months at 4 °C and can be loaded with multiple antigens (Figure 3).

### 3.4. Stimuli-Responsive Polymersomes

Tailoring of polymer structural properties enables the construction of stimuli-sensitive polymersomes in order to regulate the payload release upon exposure to various internal/physiological or external stimuli (Figure 1d). Furthermore, dual-stimuli-responsive polymersomes can be constructed, such as photo- and redox-responsive polymersomes [65], temperature- and pH-responsive polymersomes [7], or light- and reduction-responsive polymersomes [4] able to respond to two types of stimuli in order to improve the efficiency of treatment. A variety of polymersomes comprised of copolymers with characteristics that influence the response to stimuli has been described so far, resulting in pH-, temperature-, enzyme-, oxygen-, light-, reduction-, or hypoxia-responsive polymersomes, as described below. There are infinite approaches to take advantage of the characteristics of different extra- and intracellular disease environments and apply the stimulus for targeted release of drugs from the vesicles. Additionally, they can be combined with other functionalities already discussed in this section [66].

Taking advantage of the fact that reactive oxygen species (ROS) are overexpressed in a variety of diseases, ROS-responsive polymersomes could be promising diagnostic and treatment tools. Inspired by physiological mechanisms that control the permeability of cell and organelle membranes, Zhen et al. applied an innovative concept to prepare synthetic polypeptide polymersomes composed of PEGylated poly(L-cysteine) connected to hydrophobic cholesterol moieties via thioether bridges [67]. Upon exposure to oxygen species, a transition from β sheet to α helix occurs and results in vesicle thinning, enabling the release of an encapsulated load. In the case of incorporated hydrophobic compounds, increased hydrophilicity of oxidized sulfur species and modified crystallinity of the cholesterol layer decrease interactions with the load and promote the release. On the other hand, the changes in the pattern of hydrogen bonds are likely responsible for the transport of hydrophilic species captured within the hydrophilic cavity. The ability of such nanoreactors to deliver therapeutic proteins was explored via the incubation of insulin- and glucose oxidase(GO)-loaded polymersomes with glucose. H_2_O_2_ produced by the reaction of GO and diffused glucose triggered vesicle permeability and the release of functional insulin (Figure 4). 

Biodistribution studies have revealed slower clearance of encapsulated insulin (>24 h) when compared to free insulin (4 h) in diabetic mice, but faster than in normoglycemic animals, confirming in vitro observations of glucose-triggered insulin release. Entrapped insulin was also able to maintain prolonged normal glucose levels when compared with free insulin (4 h vs. 1 h, respectively), as well as higher blood insulin levels.

Reduction-responsive polymersomes have also been described. Taking advantage of the presence of glutathione (GSH) in the cytosol, the release of encapsulated cargos from GSH-responsive polymersomes can be triggered. It can be achieved by the presence of disulfide linkage in the copolymers, as they break down in response to reducing agents [68]. Alternatively, GSH-sensitive disulfide bridges can be used for the conjugation of drugs [69,70] or sulfur-containing prodrugs for the delivery of nitric oxide [71]. Recently, disulfide moieties were used to crosslink hydrophobic layers of polymersomes and enhance drug release after intracellular degradation [72].

A novel vesicular system described by Cheng et al. exploited the heterogeneous distribution of redox potential (ROS_extracellular_ 0.1–1 mM vs. ROS_intracellular_ 1–10 μM and GSH_extracellular_ 2–20 μM vs. GSH_intracellular_ 10 mM) for sequential delivery of hydrophilic (Dox.HCl) and hydrophobic (paclitaxel Ptx) drugs to cancer cells [73]. Extracellular release of hydrophilic drugs occurs via the oxidation of cystine bridges that connect the PEG and PCL blocks of the copolymer. The resulting sulfoxide and sulphone groups change the hydrophilicity and reactivity of the cystine bridges and make them more susceptible to subsequent degradation by intracellular GSH, leading to Pxt release.

Associating light and reduction-responsive block copolymers to PEG, such as poly-coumarin-based disulfide-containing monomer (PCSSMA), gave rise to a dual-stimuli-responsive PEG-b-PCSSMA-based polymersome, where disulfide linkage shows reduction-responsive characteristics and coumarin groups are photo-responsive [5]. Upon irradiation with visible light (430 nm), the generation of reactive primary amine groups cross-linked the membranes due to the amidation reactions, leading to the cleavage of coumarin moieties. The polarity reversal in the membrane allowed the release of encapsulated small molecules. The bigger molecules were released later, upon disassembly of the cross-linked vesicles provoked by incubation with GSH, which is abundant in the cytosol of cancer cells. In this way, these dual-responsive polymersomes could sequentially release different therapeutic agents.

Ph-responsive polymersomes are promising as drug delivery systems due to their potential to release the drug at the desired site depending on pH value differences [74]. Their membranes can change the conformation between swelling and shrinking according to the acidic and normal physiological pH, respectively. While protection of cargo is provided under normal physiological pH, the control of the drug delivery in an acidic tumor environment and increased selectivity of the drug action are enabled at lower pH characteristics of the tumor microenvironment [74]. Furthermore, at certain pH values where polymersomes are only semi-swollen, it is possible to reach a retarded release profile [12]. 

A hydrophobic component that is often used is a pH-sensitive poly(2-(diisopropylamino)ethyl methacrylate) (PDPA), and it was concluded that PEG-PDPA polymersomes with Dox significantly reduced drug toxicity and improved cancer therapy in a zebrafish embryo tumor model [75]. 

The assembly of a new type of polyphosphazene polymersome with a pH-sensitive ortho ester group was assisted by π–π interactions of benzene rings and Dox [76].

On the other hand, temperature-responsive polymersomes are susceptible to elevated temperature in a tumor environment. Increased tumor metabolism or externally controlled temperature can trigger drug release when polymer blocks with appropriate critical solution temperature (LCST) are used [13]. Vesicles composed of biocompatible triblock polymer poly(*N*-vinylcaprolactam)_n_-*b*-poly(dimethylsiloxane)_m_-*b*-poly(*N*-vinylcaprolactam)_n_-(PVCL_n_-PDMS_m_-PVCL_n_) experience reversible decrease in size and membrane thickness in the range of 37–42 °C due to favorable LCST of the PVCL block [77]. Sustained Dox release under physiological tumor conditions (37–42 °C and low pH) was assisted by the presence of pH-sensitive linkers. Similar effects were observed in the case of mPEG-*b*-PNIPAM-*b*-P(DEAEMA-*co*-BMA) triblock copolymer with thermosensitive poly(*N*-isopropylacrylamide) (PNIPAM) and pH-sensitive hydrophobic poly[2-(diethylamino)ethylmethacrylate (PDEAEMA) segments [7]. 2-hydroxy-4-(methacryloyloxy)benzophenone] (BMA) represents a photo-crosslinker for crosslinking the hydrophobic core for increased systemic stability [7]. Hydrophilic Dox and hydrophobic Ptx were encapsulated in respective polymersome compartments and were released in a controlled manner under tumor conditions (42 °C and pH < 7). 

Considering the overexpression of enzymes involved in pathological conditions, enzyme-responsive drug delivery systems can be very promising in cancer and infection therapies because their responsivity is very selective against specific targets. The substrate of the enzyme is integrated into the polymer and, thus, when enzymes cleave the substrate, the polymersome structure destabilizes, allowing the release of the encapsulated drug. The presence of enzymes at specific sites is of particular interest, causing drugs to be delivered only to a specific location. For example, PEG-PLA bridged with a synthetic polypeptide PVGLIG sensitive to matrix metalloproteinase 2 (MMP2) overexpressed in tumor extracellular matrix exhibited a seven-fold increase in SN38 release compared to the control formulation [24]. 

In a study by Yao et al., the site-specific photodynamic (PDT) and imaging potential of polyacrylate-based polymersomes was triggered by NAD(P)H:quinone oxidoreductase isozyme 1 (NQO1) endogenous in some cancer cells [78]. Polymer functionalized with quinone trimethyl locks was conjugated with photosensitizers Nile blue and coumarin. Without the stimuli, the theranostic function was locked by aggregation and quinone-photoinduced electron transfer quenching. After cellular uptake and NQO1 exposure, self-immolative photosensitizer release and activation were induced. An additional functional element, targeting the ligand cRGD, increased cellular internalization and contributed to the overall in vitro and in vivo cytotoxic effects upon near-infrared (NIR) irradiation of Nile blue vesicles (~90% tumor growth inhibition). In polymersomes conjugated with a nuclear magnetic resonance (MR) imaging agent, gadolinium-tetraazacyclododecanetetra acetic acid (Gd-DOTA), the moieties liberated after self-immolative quinone release led to intra/inter-polymer cyclization followed by the release of Gd-DOTA. When Gd vesicles were loaded with hydrophobic camptothecin (Cpt) and hydrophilic Dox.HCl, a dramatic increase in drug release was observed upon NQO1 exposure (>90% against <20% without stimulus). Both Nile blue and Gd-DOTA polymersomes were successfully applied for tumor imaging in vivo. 

The release of photosensitizer pyropheophorbide a (Ppa) and lapatinib (Lap) from poly(oligo(ethylene glycol) methyl ether methacrylate) (pOEGMA) polymersomes is another example of enzyme-sensitive vesicles [79]. Here, Ppa and pOEGMA were bridged by a cathepsin B-sensitive peptide linker. Embedding of dendritic Ppa into the cell membrane facilitates cell and tumor penetrability, and ROS produced by laser irradiation combined with Lap to generate synergistic in vivo antitumor effects. 

Hypoxia-responsive polymersomes can also be used following the same principle. Considering that treatment of triple-negative breast cancer (TNBC) is challenging because of rapid cell proliferation and inadequate blood flow which decreases oxygen concentration at the tumor site, it is possible to release the drug at the exact location of the tumor by constructing a hypoxia-responsive polymersome. PLA-PEG polymer with diazobenzene linker reduced under hypoxic conditions, leading to the disintegration of the polymersome membrane [62]. At the same time, the iRGD peptide integrated into the membrane directed them to penetrate solid tumors, as iRGD interacts with specific integrins present in endothelial cancer cells and promotes transcytosis extravasation [62]. 

In light-responsive polymersomes, the release of encapsulated drugs is achieved by exposing polymersomes to a photo source with a determined wavelength, allowing localized and non-invasive treatment in a specified period of time [80]. This approach enables intensive control and direction of the treatments. 

The addition of hydrophobic tetraphenylethene stilbene-type motifs (TPE) to the hydrophobic cholesterol core into PEG-TPE-Chol enables the formation of membrane pores [48]. The shape of the self-assembled aggregates based on PEG-TPE-Chol depends on the configuration of the used TPE copolymer [48]. For example, when trans-configuration was used, spherical vesicles were formed, while the cis-configuration of the same block copolymer yielded cylindrical micelles. However, the mixture of PEG-TPE-Chol isomers (trans/cis = 60/40) resulted in vesicles with porous membranes (Figure 5) during the self-assembly in water. Moreover, trans−cis photoactivated isomerization can occur under UV illumination and can make the self-assembly photo-responsive. Under high-intensity UV light, vesicles and micelles (formed by trans-PEG-TPE-Chol and cis-PEG-TPE-Chol) tend to form porous vesicles [48]. 

Functionalities can be obtained by incorporating other types of nanoparticles, such as NIR-sensitive nanoparticles (NPs). Using poly(ε-caprolactone)-poly(methacrylic acid) diblock copolymer (PCL-PMAA) with a UV-sensitive ONB (*o*-nitrobenzyl ester) and GSH-responsive disulfide linkers, photo- and redox-responsive polymersomes able to carry upconversion nanoparticles (inorganic nano-crystals with photoactivation technology) or anticancer drugs were constructed [65]. These upconversion nanoparticles allow the polymers unable to absorb near-infrared (NIR) light to upconvert NIR light into UV light and degrade the ONB linkage. PCL-ONB-SS-PMAA-based polymersome encapsulating Dox (in the hydrophilic core) and upconversion nanoparticles (in the hydrophobic bilayer) undergo a marked increase in DOX release using dual stimuli response (GSH and laser light), relative to therapeutic molecules alone [65]. Au nanoparticles (NP) can also serve as NIR-sensitive elements that facilitate polymersome degradation and can be used as building blocks of highly functionalized vesicles [81].

The overview of the recently employed copolymers presented in this section represents only a portion of copolymers that have been developed for potential use in API delivery. However, they amply demonstrate the available options for constructing polymersomes with the most diverse and tunable characteristics and properties to be used as drug delivery systems.

## 4. Preparation Methods

The self-assembly process results from the capacity of block copolymers to form organized structures in the presence of an aqueous medium, as evidenced in Figure 4. To understand the process, a series of factors need to be considered. The vesicles are formed by the noncovalent interactions between hydrophobic moieties and minimal hydrophilic interactions among hydrophilic parts (hydrophobic effect), leading to thermodynamic stabilization between the nonpolar groups and water [27,82]. These attractive and repulsive forces lead to variations in the interfacial area and determine the shape of the structures [83]. Conditions of pH, temperature, and salt concentration impact the hydrophobic interactions and, consequently, polymer organization [82]. These production parameters will depend on copolymer characteristics and can be influenced by the size, length of the chain, hydrophilic/hydrophobic ratio, charge, molecular weight, and composition of block copolymers, and also by the concentration of the solution [82]. Additionally, the nature of API also plays a crucial role in the choice of production method. 

There are several methods of polymersome preparation, and their advantages and disadvantages are represented in Table A2 (Appendix A). Beyond the described methods, UV crosslinking is an additional process in which already assembled polymersomes [12,37,84] can be cross-linked by UV irradiation to increase the structure stability or modulate the membrane permeability through pH changes [12,37].

Another type of polymer shell preparation is based on the consecutive layering of charged polyelectrolyte polymers onto a template which is later degraded, but this preparation method will not be discussed in this review [85]. 

### 4.1. Direct Hydration

In this method, an organic phase is created by dissolving block copolymers in an adequate organic solvent. After that, an aqueous solution is gradually added to the mixture, and polymers are dispersed in the aqueous phase, resulting in self-assembled structures [86]. An organic solvent can be poly(ethylene glycol) dimethyl ether with 500 Da (PEG 500 DME), which yields polymersomes with high encapsulation efficiencies [87], or oligo(ethylene glycol) (OEG) [8], a nontoxic solvent with low molecular weight [88]. This process can occur under certain conditions, such as stirring [8], controlled salt concentrations [8], and temperature [87], and in a defined period of time, in order to improve the speed, smoothness, efficiency, and robustness of the process. It is based on the dissolution of a high concentration of block copolymers in a low-molecular-weight PEG. At the melting point of the block copolymers, they will start to mix with PEG to obtain a homogeneous solution. After that, water is gradually added to dilute the low-molecular-weight PEG, resulting in polymersomes dispersed in the solution [87]. In the case of PEG_2k_-*b*-PCL_3.6k_, encapsulation of hydrophobic Ptx was achieved via the addition of Ptx into a polymer phase (encapsulation efficiency EE 40%, drug loading capacity DL 2%) [86]. Encapsulation led to an increase in size (from 118 to 137 nm), and Ptx clusters were observed within the hydrophobic layer as a possible consequence of the hydrophobicity difference between Ptc and PCL. It is presumed that during the water addition step, Ptx forms aggregates which are incorporated. When a thin layer method was applied to the system, larger polymersomes were obtained (<600 nm), but without visible Ptx aggregates. 

### 4.2. Thin Film Hydration

In this method, a dried thin film of amphiphilic molecules, formed by solvent evaporation [6], is hydrated under mechanical agitation, during which the material swells and detaches the polymersomes from the surface [18]. However, this method presents some disadvantages relative to time, cost, and lack of facilities [87]. Moreover, this method can be complemented with an extrusion step, whose advantage is the formation of polymersomes with uniform size [1]. Polymersomes obtained via thin film hydration followed by extrusion are smaller than those produced by nanoprecipitation, probably because vesicles were forced through the small pores of a membrane during extrusion [13]. 

### 4.3. Electroformation

This method derives from the thin film hydration technique as it involves the swelling of an amphiphilic block copolymer [18]. It yields larger vesicles (up to a micrometer) and allows reasonable control of size distribution. However, the separation of the block copolymers from the surface occurs through an applied alternating current. The mechanical stresses resulting from electro-osmosis create vibrations that match the frequency of the alternating electric field, which is oriented perpendicular to the electrode. These vibrations cause bilayers to detach from the film that has been placed on the electrode. Subsequently, the water can permeate through the block copolymer, exposing the hydrophobic moieties to water. This exposure drives the process of self-organization, leading to the formation of polymersomes.

### 4.4. pH Switch Method

The pH switch method takes advantage of the pH-responsive characteristic of copolymers to self-assemble them into vesicles under aqueous conditions [89]. It can be started by increasing the initial solution pH to approximately neutral through the addition of sodium hydroxide (NaOH). When pH values are lower than the pKa of copolymers, the protonated state prevails, and polymers are soluble. When pH is higher than pKa, copolymers deprotonate and become water-insoluble. In this case, hydrophobic blocks come together allowing the formation of polymersomes. Mechanistically, when the protonation is low, the hydrophobic chains collapse and have low volume, and hydrophobic interactions drive the formation of micelles. As they become more protonated, the hydrophobic chains swell, and the volume of the hydrophobic segment increases. It shifts the packing parameter, and polymersomes are formed. The morphology of self-assembled structures depends on the temperature, which regulates pKa values and, consequently the molecular packing factor [90].

### 4.5. Solvent Displacement Method (Nanoprecipitation)

Owing to its simplicity and ability to enable control of morphology, nanoprecipitation has become a widely used method of polymersome fabrication [91]. It is based on the use of organic solvents (or their mixture) and aqueous non-solvents that are fully miscible at all proportions. In a simple experimental setup, water is typically added to the organic solution. As a result of the increased hydrophilicity of the mixture, a phase separation of the hydrophobic block occurs and consequently leads to a self-assembly process [9,37]. To achieve pure self-assembly phases, a gradual addition of water coupled with moderate stirring proved to be indispensable [43]. In the end, the organic solvent (which can be DMSO or THF) is removed, for example, via extensive dialysis against an aqueous medium [9,43]. This method allows the formation of more stable and well-defined structures in pure phases relative to the film rehydration method [43]. Various studies report that the formation of spheric polymersomes occurs at a specific ratio of non-solvent/solvent, typically up to 50% *v*/*v* [9,92,93]. During the slow addition of the aqueous phase via a syringe pump (e.g., 1 mL/h), local supersaturation leads to the creation of nuclei that further aggregate or grow to form polymersomes [94]. Although the use of a syringe pump provides flux rate control, manual precipitation can also be used. In the case of poly(ethylene glycol)-*b*-poly(*N*,*N*-diethylaminoethyl methacrylate) (PEG_n_-*b*-PDEAEM_m_) polymersomes, an aqueous phase (final concentration 90% *v*/*v*) was added to the polymer solution in THF dropwise. On the contrary, polymersomes loaded with Au nanorods were prepared by inverse addition of polymer solution (DMSO) to the nanorod dispersion in water, precluding definitive conclusions on the influence of processing parameters on polymersome size [95]. However, the influence of the order in which the solvents are mixed on the size was demonstrated in the case of PEG-PTMC polymersomes [28]. To prepare Dox-loaded PLA-based polymersomes, a solution of polymer and Dox was added slowly into PBS to prevent polymer precipitation [96]. Although polymersomes can be loaded during nanoprecipitation by dissolving API in an appropriate solvent phase, polymersomes can be previously prepared, lyophilized, and loaded during rehydration with an aqueous solution. This approach was used for loading hyaluronic-based HA-PLA polymersomes with the enzyme β-galactosidase [97]. 

Although polymersomes with relatively small and uniform sizes can be prepared in laboratory conditions with the experimental setups described in the above examples, fast nucleation and precipitation during supersaturation are crucial for the formation of uniform nanoscaled particles. The need to limit the nucleation step led to the development of flash nanoprecipitation techniques in which the mixing rate and contact time of solvent and non-solvent can be minimized (milli to microseconds) [98]. Fine-tuning of processing parameters like reservoir volume, polymer concentration, solvent nature, and temperature can further be used for the optimization of polymersome properties. Fast removal of organic solvent or its dilution stabilizes the obtained nanostructure. Flesh nanoprecipitation has only recently been applied in polymersome fabrication [99]. Poly(propylene sulfide) PEG_17_-*b*-PPS_30_ has been mixed in confined impingement jets (CIJ) mixers. Multiple impingements through the system resulted in single bilayer polymersomes with the same structural characteristics (PDI < 0.15) as in those obtained by the thin film method and time-consuming extrusion, but without the loss of material. The method is especially adaptable to the loading of large polar biomolecules with preservation of biological activity, and the encapsulation efficacy (EE 16–43%) surpasses that of other conventional methods [99]. At the same time, high EE was achieved for hydrophobic molecules (up to 98%). The cargo was dissolved in the appropriate solvent phase before mixing. Additionally, the process can be scaled up and used for the production of reproducible large-scale batches needed for clinical application (30 mL with 20 mg/mL polymer) [100]. 

### 4.6. Single and Double Emulsion Method

The single emulsion method starts with the preparation of an aqueous solution (aqueous phase) of a therapeutic cargo and an organic solution containing block copolymers (organic phase) [24]. After that, the aqueous solution is mixed with an immiscible organic solvent compatible with the polymer, forming a primary emulsion with tiny droplets in the organic phase. A surfactant can be added in order to prevent the coalescence of the emulsion. Finally, the organic solvent is removed, and copolymers self-assemble around the aqueous droplets, yielding polymersomes [24,65]. The double emulsion method involves an additional step that includes two emulsifications in the process. Upon stabilization, the primary emulsion is added to a hydrophobic organic phase creating a double emulsion. Only then is the solvent removed and the polymersomes formed [65]. This method shows disadvantages associated with uniformity of size. However, these can be overcome by optimizing the copolymer characteristics and process conditions [84]. Additionally, this method demonstrates good encapsulation efficiencies [24,65]. For example, the double emulsion method was used to load poly(ε-caprolactone)-ONB-SS-poly(methacrylic acid) (PCL-ONB-SS-PMAA) polymersomes with upconversion nanoparticles (UCNP) and Dox. When polyvinyl alcohol (PVA) solution was added to the UCNP and copolymer solution in CH_2_Cl_2_ under sonication, UCNP (EE 8.7%) was embedded into the hydrophobic shell. This first emulsion was subsequently added to another PVA solution. To achieve doubly loaded polymersomes, the Dox (EE 85%) solution in DMSO was added to the PVA (aqueous) solution during the preparation of the first emulsion [65]. 

### 4.7. Microfluidics Synthesis

Microfluidics emerges as a sophisticated technology applicable to small volumes of solutions within microscale fluidic devices, offering precise and controlled process manipulation [101]. Notably, the self-assembly process heavily depends on both external (temperature) and internal parameters (such as interfacial and viscous forces). The self-assembly process can be effectively managed by controlling fluid viscosity and mass density through precise flow rates of the solvent and anti-solvent in microchannels. In this regard, microfluidics represents potentially reliable and scalable technology for controlling the self-assembly of polymeric nanoparticles [101]. Another advantage is that microfluidics demonstrates its potential in scaling up polymer-based nanomedicine production due to its continuous flow operational process, a significant factor when transitioning to clinical trial applications. A recent study has indeed confirmed that controlling nanoprecipitation conditions (flow rate, water content, and polymer concentration) via microfluidics allows fine-tuning and controlling the size and polydispersity index of nanoparticles while maintaining their shape with impeccable batch-to-batch reproducibility [28]. 

The small dimensions of microfluidic channels provide a higher surface-to-volume ratio, reduced diffusion times, kinetically controlled nanoprecipitation process as well as control of morphology, as demonstrated in the example of poly(ethylene glycol)-*b*-poly(*N*-2-benzoyloxypropyl methacrylamide). Both micelle and polymersome formation were observed by mixing THF polymer solution with water [101]. Microfluidic mixers with different setups can be used to control the flow and mixing speed of separate fluid streams. Nonetheless, comparative studies of different mixer setups in polymersome fabrication are rare. In a recent study by Martin et al., the influence of flow subdivision (micromixer) and multiple chaotic flows (herringbone mixer) were compared on the example of PEG-*b*-PTMC copolymer. Although both mixers performed similarly under the same conditions, it was possible to identify the critical water content (20% DMSO in PBS) [102]. A micromixer chip was used to prepare thermosensitive poly([*N*-(2-hydroxypropyl)]methacrylamide)-*b*-poly[2-(diisopropylamino)ethyl methacrylate] (PHPMA_35_-*b*-PDPA_75_) nanocarriers for Dox delivery by optimizing the flow rates and ratios of the organic and PBS phases (~60 nm, PDI 0.06, with an encapsulation efficacy of 53.1% and loading capacity of 9.8%) [103]. Loaded polymersomes were prepared by co-dissolving Dox and polymer in the organic phase. This method demonstrated advantages over the thin film method where larger and less monodispersed systems were observed (100 nm, PDI 0.15). 

An alternative experimental approach is based on emulsification by mixing two immiscible fluids. Here, the mechanism of droplet formation, and their size, polydispersity, and shape are defined by flow rate, viscosity, and interfacial tension. This method is especially interesting for the formation of artificial cells and organelles since high-order vesicles can be formed (vesicles within vesicles or concentric shells). Apart from linear configurations (see the above examples), non-planar and capillary devices can be used in which the carrier phase envelops the dispersed phase [104,105]. 

### 4.8. Polymerization-Induced Self-Assembly

Polymerization-induced self-assembly (PISA) combines copolymer synthesis and self-assembly steps, thereby avoiding multiple processing steps. During the PISA, the solvophilic polymer chain is extended by solvophobic monomer polymerization. After a critical hydrophobic chain length is reached, self-aggregation commences. Control of hydrophobic chain extension can provide different aggregate architectures. A variety of chain-propagation mechanisms is available and can be matched to the chosen polymer monomer, reaction conditions, application site, and therapeutic cargo. In PISA, in situ incorporation of APIs into different polymersome compartments is possible via covalent (via conjugation to polymer chain or monomers) and non-covalent interaction. On the other hand, drug encapsulation can be performed after the vesicle assembly process [106]. A detailed overview and recent developments in the application of this method for API-loaded nanosystems have been described elsewhere [107,108]. Here, some recent examples will be highlighted.

The degree of polymerization and temperature had a marked influence on a series of poly(glycerol monomethacrylate)-poly(2-hydroxypropyl methacrylate) PGMA_62_-PHPMA_600–1400_ polymersomes prepared by photoinduced PISA during aqueous reversible addition–fragmentation chain transfer (RAFT) polymerization [109]. Mechanisms involved transitions in membrane thickness or shape (from spherical to donut and tubular polymersomes). 

The choice of polymersome components can have a decisive impact on the sensitivity to stimuli [110]. Recently, H_2_O_2_-sensitive polymersomes were prepared by RAFT-extension of hydrophilic poly(*N*-2-hydroxypropyl methacrylamide) P(HPMAm) with *N*-(2-(methylthio)ethyl)acrylamide (MTEAM) monomer [110]. P(HPMAm) was derivatized with folic acid, a targeting ligand for cancer cells, prior to RAFT chain extension. Additionally, hydrophobic and hydrophilic model compounds were successfully incorporated during PISA (hydrophobic Nile red EE 72%, loading capacity LD 0.04%, hydrophilic Cy5 EE 23%, and LD 0.05%). Unlike when bulkier hydrophilic poly(ethylene glycol) methacrylate was used in combination with P(MTEAM), P(HPMAm)_43_-*b*-P(MTEAM)_300_ was efficiently degraded by 10mM H_2_O_2_ [111]. Recently, seeded photoinitiated PISA was used for the construction of vesicles with tunable membrane thickness and composition under mild conditions [112].

### 4.9. Other Methods

Although the introduction of flow-based techniques like flash nanoprecipitation and microfluidics introduced improvements in size and shape control over batch fabrication, they often rely on the use of complex and expensive equipment, with limited scalability and ability to apply downstream polymersome manipulation. To compensate for these drawbacks, Wong et al. designed an affordable continuous two-flow system that enables downstream size and shape control, with a high production rate (>3 g/h), applicability to a wide variety of polymers, and setup modularity [113]. By adjusting the ratio of solvent/non-solvent and their respective flow rates it is possible to control nanoaggregate shape (0.7 for PEO_44_-*b*-PS_86_). Additionally, the metastable nature of obtained polymersomes enabled the trapping of populations with the desired size by removing organic solvents during maturation. The process was robust toward increases in polymer concentration and flow rate (up to 9 mg/mL and 8 mL/min). The process was also applicable to polymers with modified polarity (PEO_44_-*b*-P4VP_21_-*b*-PS_300_ with an additional hydrophilic poly(4-vinylpyridine) (P4VP) linker and PAA_26_-*b*-PS_81_ with a protonable hydrophilic polyacrylic acid chain). To provide downstream processing, annealing (20–70 °C for control of polymersome size equivalent to the aging process), and cooling (size equilibration) loops followed by an additional pump (organic solvent removal and shape modification) were added. 

## 5. Recent Advancements in Polymersomes as Drug Delivery Systems

The use of polymersomes as drug delivery systems is promising because it takes advantage of their tunable stability, selective permeability, sustained drug release, and targeted delivery [114]. Therefore, keeping in mind the characteristics of polymersomes, they can be used in a variety of applications, and some of them will be described below. 

### 5.1. Chemotherapy

Chemotherapeutic agents are associated with poor efficacy against cancer cells due to low systemic stability and resistance development but also with off-target toxic effects. Incorporating them into vehicles can improve the pharmacokinetic profile of systemically administered drugs and direct them to the specific sites of action, overcoming adverse side effects in the process [75]. 

For example, Dox leads to the progressive development of heart failure, causing irreversible damage in the cardiac muscle and dysregulation of the immune system in dose- and treatment-time-dependent ways [42]. The photosensitive *o*-nitrobenzyl (ONB) linker was combined with redox-sensitive disulfide in poly(ε-caprolactone)-ONB-SS-poly(methacrylic acid) (PCL-ONB-SS-PMAA) [65]. Polymersomes loaded with UCNP and Dox completely disintegrated after laser irradiation (30 min at 980 nm), while exposure to 5 mM GSH led to decomposition within 15 min. Cumulative effects of GSH and irradiation on Dox release were observed. Synergistic anticancer effects were observed in vitro and in vivo after irradiation. Additionally, UCNP luminescence was also applied for imaging in nude mice bearing the A549 lung tumor xenograft. A biodistribution study showed that accumulation in tumor and liver gradually declines after the first day, but is still present four weeks later.

An example of a promising therapeutic application of polymersomes was observed in colorectal cancer, one of the most fatal cancers worldwide [115]. 7-ethyl-10-hydroxy camptothecin (SN38), which is a more potent active metabolite of the well-known irinotecan (Topoisomerase-1 inhibitor), has compromised clinical application due to low solubility and poor stability, as it is converted into its inactive form under physiological pH. Enzyme-responsive polymersomes can be promising options for targeted drug release when exposed to enzymes that are overexpressed in particular cancer tissues. PEG-PLA linked through a cleavable synthetic peptide sequence, PVGLIG, were prepared for targeting the tumor-associated MMP-2 enzyme, allowing a seven-fold higher release rate than when exposed to the MMP-2 enzyme at physiological conditions [24]. On the other hand, the addition of targeting DNA AS1411 aptamers yielded a guided drug delivery against nucleolin-positive cells. The in vitro study demonstrated an increased cytotoxicity of Apt-SN38-peptide-polymersomes against C26 cancer cells. A PEGylated surface that hampers the detection of nanoparticles by the mononuclear phagocyte system (MPS) and prolongs their half-life time in the bloodstream, as well as their small size (<200 nm) enables their uptake via the EPR effect. Their more specific and efficient drug delivery allowed increased tumor accumulation and penetration, enhanced cellular uptake via receptor-mediated endocytosis, and presented a promising tool to improve the efficacy and safety of cancer chemotherapy. Moreover, in vivo studies also demonstrated high therapeutic rates for polymersomes with cleavable peptide sequences and even higher rates for polymersomes with targeting ligands on their surface. 

In some cases, self-aggregation into polymersome structures can involve using organic solvents or increased temperatures incompatible with biologicals. The copolymer poly(3-methyl-*N*-vinylcaprolactam)-*b*-poly(*N*-vinylpyrrolidone) (PMVC_58_-PVPON_65_), composed of hydrophilic polyvinyl derivatives assembles in water at decreased temperatures (T < 20 °C) [42]. Control over lower critical solution temperature (LCST) and aggregation properties was obtained via hydrophobic modifications of the PVPON component and copolymer composition. The model drug (Dox) was loaded at room temperature without the use of organic solvents with exceptional efficiency (EE 95%, LC 49%, 360 nm). Another highlight of this copolymer system is its resiliency against physiological conditions, and very low Dox release was observed at pH 7.4 and 5 during 24 h. Moreover, the presence of poly(vinylpyrrolidone) in the outer polymersome corona increased the hydrophilicity of the vesicles resulting in a good in vitro stability in serum. The in vivo stability of PMVC_58_-PVPON_65_ and its ability to attenuate Dox toxicity was evaluated in vivo by administering a toxic non-survival dose to mice (15 mg kg^−1^ week^−1^) and was compared to liposomal Dox. Although the authors did not provide the liposomal composition, both formulations provided 14-day survival, albeit some toxicity was observed with liposomal Dox. The mild encapsulation conditions make this formulation promising for the delivery of sensitive chemo and biological therapeutics. Nonetheless, the introduction of stimuli-sensitive motives will be needed to enable the release of therapeutics in order to broaden the therapeutic applicability.

In another recent example, in the Ptx-loaded PEG-PCL polymersomes, more attenuated and prolonged release of the drug was observed under pH5 than at pH7, presumably due to more gradual hydrolysis degradation of the polymer, but also with the possible contribution of aggregated state of Ptx within the PCL layer [86]. In an orthotopic syngeneic animal model of glioblastoma tumor, polymersome Ptx demonstrated greater tumor decrease and eradication than free Ptx or clinically approved liposomal formulation (Lipusu^®^). 

Bacterial keratitis or acute ophthalmic infection is another disease characterized by elevated ROS levels, and alleviation of oxidative stress can be combined with antibiotics to mediate the infection. Block copolymer poly(polyethylene glycol methyl ether methacrylate)-*co*-*N*-benzylacrylamide)-*block*-poly(2-methylthioethyl methacrylate) (P(PEGMA_10_-*co*-PBA2)-*b*-PMTEMA_25_) with ROS scavenging thioether fragments was combined with mucoadhesive and bacteria-targeting phenylboronic acid motives for in vivo delivery of antibiotic ciprofloxacin hydrochloride [116].

Combination therapy is based on concurrent targeting of different disease mechanisms by using two or more chemotherapeutic agents [117]. The improved therapeutic efficacy results from reduced required dosage of different drugs compared to monotherapy, decreased risk of side effects and drug resistance associated with therapy [26,118]. The concept was used for targeted co-delivery of Dox and camptothecin (Cpt) to non-small-cell lung cancer (NSCLC) by hyaluronic acid-*b*-polycaprolactone polymersomes [26]. Hyaluronic acid (HA) is biocompatible and biodegradable, and can be used as a targeting agent due to interactions with tumor receptors that are overexpressed in many tumors of epithelial origin. To improve accumulation in cancer cells, it was conjugated with a FOXM1 DNA aptamer. Drugs were loaded with high efficiency in both mono and co-loaded monodispersed polymersomes (140–170 nm; Dox EE 80% and Cpt EE 55%, LC 4–6% for all formulations). It was demonstrated that 24% of Dox and Cpt were immediately released and the rest was released in a sustained manner. The ability of HA to target CD44 receptors resulted in cellular uptake by endocytosis and was complemented by the ability of FOXM1 to downregulate efflux pumps. Increased cellular accumulation of drugs exerted synergistic in vitro activity against NSCLC by inducing apoptosis. Moreover, in vivo tests in the heterogeneous heterotopic SK-MES-1 murine model showed a significant tumor suppression effect [26]. 

Hydrophilic Dox and hydrophobic Ptx encapsulated in mPEG-*b*-PNIPAM-*b*-P(DEAEMA-co-BMA) polymersomes demonstrated efficient cellular uptake and a synergistic cytotoxic effect in human cervical HeLa and breast MCF-7 cancer cells [7]. The L929 cell line derived from normal subcutaneous tissue was used as a model of healthy cells. The absence of a cytotoxic effect demonstrated a lack of drug release under normal physiological conditions (pH 7.4 and 37 °C), confirming the sensitivity of the copolymer to the characteristic cancer cell microenvironment (pH 6.0–7.0 and 42 °C). Synergistic effects were also observed for combining these two drugs delivered by Pluronic FA-F127-PLGA/PLGA-F127-PLGA polymersomes to ovary carcinoma OVCAR-3 cells [119].

In another recent example of combination therapy, PLA-HA polymersome delivered Dox with peptide melittin. In multidrug-resistant MCF-7 cells, melittin suppressed efflux pumps and showed synergistic cytotoxic effects in combination with Dox [120]. 

### 5.2. Immunotherapy

Immunotherapy has been studied as a weapon for treating or retarding the progression of tumors, being used for long-term curative effects [25]. The use of polymersomes in vaccine formulations brings new perspectives on antigen protection, controlled drug release, elevated antigen density, and co-delivery of different components to the same cell [20].

Decorating polymersomes with Fc fragment and CpG ODN (CpG oligodeoxynucleotide is an example of a potential vaccine, and has already been discussed in Section 3.3 [20]. Other studies have investigated the potential use of polymersomes in antigen vaccines for tumor immunotherapy [121,122,123]. Tumor-specific antigens or tumor-associated antigens, which can be proteins or peptides, are processed by antigen-presenting cells (APCs), and activate T cell-mediated immune responses. As their application is associated with some limitations, such as poor immunogenicity, biocompatibility, cellular uptake, and fast elimination from the bloodstream, a delivery system can also be advantageous. For that, the drug delivery system must allow adequate protein release and maintain the stability of proteins. Oncolytic peptide LTX-315 is able to induce immune responses and tumor regression. Xia et al. designed a cyclic peptide cRGD-functionalized chimeric polymersomes for delivery of the LTX-315 and the immunoadjuvant CpG to be used in immunotherapy against malignant B16F10 melanoma [25]. Polymersomes loaded with positively charged LTX-315 were assembled from PEG-P(TMC-DTC)-poly(aspartic acid) (PEG-P(TMC-DTC)-PAsp) which provided a negatively charged inner shell surface [124], and cRGD-decorated PEG-P(TMC-DTC) [125]. To encapsulate CpG, PEG-P(TMC-DTC) conjugated to positively charged spermine was used [126]. The in vivo studies demonstrated that systemic administration of cRGD-functionalized polymersomes with LTX-315 peptide, combined with the immunoadjuvant CpG-loaded polymersomes and with anti-PD-1 antibody, resulted in a strong immune response (increasing CD8+, cytotoxic T lymphocytes, helper T cells, reducing T regulatory cells in the tumor, and increasing cytokines, such as IL-6, TNF-α, and IFN-γ), and long-term immune memory protection due to an increase in memory T cells (Figure 6) [25]. Systemic instead of intratumoral delivery of functional LTX is a great improvement of this type of therapy and could be broadened to treat metastases or inaccessible tumors. 

### 5.3. Nucleic Acid Delivery

Biopharmaceuticals or biologics represent a large fraction of recently approved therapeutics composed of amino acids or nucleotides, and include hormones, antibodies, deoxyribonucleic acid (DNA), and diverse ribonucleic acids (RNAs) [27,127]. Despite their potency and specificity, some factors compromise their therapeutic efficacy, including structural complexity, poor stability, immune cell activation, low membrane permeability, and in vivo clearance, leading to low intracellular delivery and short circulation half-lives (sometimes in minutes). Nanocarriers such as polymersomes can protect them and prolong their circulation time, promote targeted delivery and cellular internalization, and improve their capacity to exert the therapeutic effect effectively and safely [27,127,128]. Nevertheless, problems related to preparation and encapsulation have been described, including insufficient loading and encapsulation efficiency, as well as drawbacks associated with the instability of biologics during the process. With respect to this, new preparation methods have emerged to overcome these limitations (Section 5). 

Non-coding RNAs like silencing siRNA are strong tools for antitumor therapy, albeit with drawbacks associated with safety, efficiency, instability, and lack of targeting capacity. Thus, polymersomes are good candidates for targeted RNA delivery [129,130].

The use of biodegradable, non-ionic copolymers in polynucleotide delivery could circumvent undesired cytotoxic contributions by the positively charged components. Poly(*N*-vinylpyrrolidone)-*b*-poly(dimethylsiloxane)-*b*-poly(*N*-vinylpyrrolidone) (PVPON_14_−PDMS_47_−PVPON_14_) is the first example of non-ionic polymersomes for siRNA delivery [131]. Polymersomes prepared by the thin film method (<105 nm) were in this case smaller (20%) than those prepared by nanoprecipitation, but this was attributed to the use of the additional extrusion step. Polymersomes loaded with PARP1 siRNA and labeled with the Cy5.5 imaging tag significantly reduced targeted protein levels and suppressed human breast cancer MDA-MB-361TR and murine 4T1 proliferation by 35% after a 6-day exposure. In a syngeneic murine model, a four-fold increase in animal survival was observed against the control animals after three months (80% vs. 20%, respectively).

In another study, cholesterol-PEI-oxidized alginate (OA) was assembled into polymersomes with a PEI corona and membrane composed of hydrophobic cholesterol and electrostatic PEI/OA complex (NP) [130]. siRNA loading was, in this case, achieved by simple mixing of two aqueous solutions, and was subsequently decorated with iRGD by physical interactions (NP/siRNA/iRGD 9:1:1 weight ratio) [63,130]. Polymersomes demonstrated deep penetration of MCF-7 spheroids and cellular uptake dependent predominantly on the ligand-induced endocytosis. NPs loaded with antiangiogenic siVEGF efficiently suppressed neovascularization in a zebrafish model. 

The compartmentalized nature of polymersomes was exploited for the co-delivery of tumor growth factor β siRNA (siTGF-β) and shikonin (SK) [132]. The combined immunotherapeutic effects of siTGF-β and SK loaded into a hybrid nanoassembly of PEI-PCL and FA-decorated 1,2-distearoyl-*sn*-glycero-3-phosphoethanolamine-*N*-polyethyleneglycol led to immunogenic cell death and infiltration of cytotoxic T lymphocytes in a syngeneic orthotopic in vivo model of triple-negative breast cancer. At the same time, the formation of lung metastasis was suppressed, and long-term antitumor memory delayed the growth of secondary tumors. 

Highly functionalized polymersomes for the delivery of retinoblastoma-binding protein 4 siRNA and temozolomide were assembled from diblock Angiopep-2-PEG-*block*-poly-(2,2,3,3-tetrafluoropropyl methacrylate) and triblock PEG-*block*-poly-(2,2,3,3-tetrafluoropropyl methacrylate)-*block*-poly[(*N*-(3-methacrylamidopropyl) guanidinium)]. Here, positively charged guanidinium moieties were used for siRNA complexation. The addition of fluorine increases the hydrophobicity and stability of the membrane and has previously been associated with increased efficacy of polyplexes [133]. Efficient crossing of BBB mediated by the targeting ligand Angiopep-2 led to a synergistic therapeutic effect in a transgenic orthotopic model of glioblastoma [134]. 

### 5.4. Protein Delivery

A similar strategy based on the derivatization of hydrophilic sides with ionizable groups promotes the encapsulation of large amphoteric/amphiphilic proteins and was already presented for the delivery of nucleic acids or immunogenic factors. Granzyme B was loaded into PEG-*b*-poly(trimethylene carbonate-*co*-dithiolane trimethylene carbonate)-*b*-spermine for multiple myeloma therapy [126]. Self-assembly by nanoprecipitation was accompanied by self-crosslinking and was followed by decoration with HA (HA-RCLP-GrB). Under reductive conditions, 80% of the protein was released within 24h. Formulation with optimized HA density (30%) induced apoptosis in vitro, and alleviated osteolysis in an orthotopic murine mode with prolonged survival time. 

HA-PLA loaded with β-galactosidase was applied for enzyme replacement therapy (ERT) in a lysosomal storage disorder, GM1 gangliosidosis. Successful uptake of polymersomes by diseased fibroblasts promoted healthy autophagic activity [97]. 

Oral delivery is the preferable non-invasive administration route from the perspective of patient adherence. However, it is not applicable for big hydrophilic molecules sensitive to the harsh environment of the digestive tract and poor epithelium adsorption. ABA copolymer PEG_5_-poly(propylene)*_6__8_*-PEG_5_ (poloxamer 401, Pluronic L-121) polymersomes have the ability to transfer large molecules like proteins across biological barriers and could be potentially used to cross endothelial intestinal barrier for systemic delivery of APIs or to treat intestinal inflammation [135]. Polymersomes loaded with the model IgG-FITC antibody (Ab) or therapeutic adalimumab were prepared by co-dissolution in PBS followed by purification by size exclusion chromatography and extrusion (~200 nm, PDI < 0.2). Loaded polymersomes demonstrated potential for intestinal permeation evaluated in a CaCo2 model and exerted anti-inflammatory activity by lowering TNFα levels in lipopolysaccharide-activated macrophages (murine J774A.1). The attenuation of TNFα levels observed at the basolateral side of a co-culture CaCo2/macrophage model points to successful release of Ab from the polymersomes, suggesting that the system could be applied for oral Ab delivery across the intestinal epithelium and subsequent release in the subepithelial compartment. However, the polymer concentration needed to deliver therapeutic levels of Ab was high enough to cause cell death and increase TNFα levels, raising the question of whether this polymer combination is the most appropriate delivery system. 

A quality-by-design approach was used recently to develop L121/catalase for topical UV damage repair [136]. Polymersomes were able to penetrate into viable epidermis and dermis after topical application and decreased lipid peroxidation upon UV irradiation, making them an interesting system for dermal therapy [137].

Although protein delivery by polymersomes is usually related to their encapsulation within the hydrophilic internal area, Geervliet et al. decorated the surface of 2-(*N*,*N*’-diethylamino)ethyl methacrylate-based polymersomes with therapeutic collagenase type 1 (matrix metalloproteinase-1 MM1) for early liver fibrosis treatment [138]. Polymersomes exhibited good pH, osmotic and shear-force stability, and preserved enzyme activity after optimization of purification and storage conditions. Limited release of MM1 from polymersomes indicates that a high degree of non-covalent interactions was established and that most of the MM1 was embedded within the polymersome membrane. When applied in vivo, MM1 vesicles inhibited collagen-I formation and inflammation, thereby alleviating early liver fibrosis symptoms. 

Most of the formulations presented in this overview were prepared for administration as individual polymersomes, but they can also be embedded within different matrices. For example, hybrid polymersome–hydrogel composites [139] and polymersome-loaded microneedle patches [140] have been described. In a recent study by Edmans et al., glycerol monomethacrylate (GMA) and hydroxypropyl methacrylate (HPMA) polymersomes loaded with F(ab) antibody fragments were electrospun into bead-on-string mucoadhesive PEO nanofibers for buccal administration [141]. Polymersomes maintained the morphology and F(ab) functionality during the electrospinning process, and efficient penetration of released polymersomes into the epithelium was observed in reconstructed human oral epithelia.

### 5.5. Photodynamic Therapy

The incorporation of porphyrin photosensitizer into the dextran backbone exploited π–π interactions to assemble photosensitive polymersomes. Dextran was additionally grafted with diazo and β-cyclodextrin (βCD) moieties, which contribute to cohesive forces via host–guest interactions. Upon UV radiation, diazo-βCD dissociates with an increase in size (from 200 to 250–300 nm) and surface morphology changes, which suggest a rearrangement of corona chains and could be used for on-demand drug release. Porphyrin photo-irradiation produces singlet oxygen, which is toxic for tumor cells [142]. 

The presence of BODIPY photosensitizer in PEG-P(CLgTMC) polymersome allows a spatiotemporal control of the treatment, and it was concluded that the presence of a photosensitizer leads to ROS production in the presence of NIR irradiation light, and results in cancer cell apoptosis. The formulation was administered intratumorally and remained localized for at least 96 h, and tumor growth inhibition was observed upon NIR light irradiation [8]. 

In another study, the photosensitizer used was rose bengal, a dianionic fluorescent dye with a potential application in anticancer therapy through the generation of singlet oxygen when exposed to irradiation [12]. However, it has some disadvantages, such as a short half-life, a tendency to form aggregates, as well as a negative charge, which hampers cellular uptake. Therefore, polymersomes can be promising in allowing its protection, increasing its solubility, and directing its transport to the exact action site. Studies in carcinoma cell lines observed a slight increase in drug uptake by the cells, probably because of an almost neutral polymersome surface charge. Moreover, increased intracellular ROS generation and higher toxicity than for free rose bengal were observed [12]. 

### 5.6. Sonodynamic Therapy

Sonodynamic therapy can be an alternative to photodynamic therapy since they act similarly but with different triggers. While photodynamic therapy depends on lasers, sonodynamic therapy depends on ultrasound [143]. The mechanism of the last one involves the activation of a sonosensitizer that can transform oxygen into ROS, and damage nucleic acids and proteins in the cells, leading to their apoptosis. The advantages over photodynamic therapy include deeper and less invasive tissue penetrability, making it suitable for use in cancers located deep in the body [144]. 

A stable PLA-PEG-based polymersome capable of loading both sonosensitizer verteporfin (Vp) and Dox (>95%) with high efficiency has been developed [144]. Upon efficient intracellular uptake of both Dox and Vp by endocytosis, efficient production of ROS was followed by increased cytotoxic effect in response to ultrasonic energy. Moreover, the accumulation of polymersomes in the tumor site via the EPR effect was confirmed, evidencing selectivity for the tumor site. Finally, these polymersomes revealed no severe cytotoxicity in the body. Combined therapy was more effective in vitro and in vivo relative to single therapy [144].

## 6. Regulatory Issues

Another important requirement for the development of nanosystems is regulatory issues. The regulatory field related to nanomedicines is emerging and efforts are being made to develop international legislation [145]. So, nanomedicines follow the rules already used for drugs without the involvement of nanotechnology, being classified in the same categories as new drugs, biopharmaceuticals, and generic products [146]. However, the physicochemical properties (such as size, shape, surface moieties, composition, mechanical properties, and material type) of nanomaterials impact the quality, efficacy, and safety of the end product, as well as the pharmacodynamic and kinetic properties of nanomaterials in the human body. In this regard, additional nanotoxicological studies are essential in order to determine undesirable effects and assure the safety of products for the human body and for the environment, with the aim of understanding how medicines interact with the tissues and what are the toxic pathways. This would allow us to highlight the interests in this area and propel the advancement of nanomedicines to clinical practice [147]. The regularity approach is particularly important in polymerization-based systems as polymersomes, due to the toxicity of residual monomers, the catalyst used for the reaction, and the possible surfactant agents used to stabilize the systems. Moreover, if polymerization involves drug loading, the reaction conditions should be guaranteed to not compromise drug stability during fabrication and storage [35]. 

At this moment, there are no established regulatory definitions relative to “nanomaterials”, “nanotechnology”, and “nanoscale”, much less related to polymersomes in particular [51]. Therefore, the present work will refer only to regulatory considerations relative to nanoparticles in general. Considering that these terms are related to materials with dimensions between approximately 1 nanometer (nm) and 100 nm, the Food and Drug Administration (FDA) issued nanotechnology considerations guidance for FDA-regulated products applicable to “products that involve the application of nanotechnology” or “nanotechnology products”, which is the case of polymersomes [148]. In this regard, as nanotechnology products show different attributes than conventional products, it is expected that examination is necessary. However, the FDA regulates them according to the specific legal standards applicable to the products under its jurisdiction. As there are no specific rules related to the regulatory framework of nanoproducts, the producers are advised to contact the FDA as early as the development process to understand the scientific and regulatory issues for their particular product [51]. In general, the FDA considers two points that should be evaluated in FDA-regulated products involving the application of nanotechnology, which are related to particle dimensions and dimension-dependent properties or phenomena. After that, aspects such as safety, effectiveness, public health impact, and regulatory status of the product should be evaluated [51]. 

The FDA also issued another guidance for human drug products including biological products containing nanomaterials, to characterize, control, test and qualify the nanomaterial components in the drug product (docket number: FDA-2017-D-0759). As they differ in chemical, physical, or biological properties relative to large-scale counterparts, they can have an impact in terms of bioavailability, blood circulation time, and distribution in the organism, influencing the quality, safety, or efficacy of drug products. On the other hand, taking into account the targeted characteristics of nanomaterials, including polymersomes, more information is needed relative to their interactions with biological systems to establish the influence of the intrinsic and extrinsic factors, and their immunogenic potential. The FDA considers that to evaluate the potential risk of a drug product containing nanomaterials, it is important to characterize the nanomaterials, understand their use and application, and their relationship with the quality, safety, and efficacy of the product. So, a detailed description and characterization such as that in Table A3 (Appendix A) is recommended, in order to determine the safety and efficacy of the drug product. For clinical studies, drug products containing nanomaterials can be classified as low-, medium-, or high-risk to indicate clinically significant changes in exposure, safety, and/or effectiveness, and the clinical studies should be designed according to this classification [149].

Beyond all these considerations, it is important to take into account other aspects: the formation of protein corona or protein fouling, and the EPR effect in the tumor environment. The protein binding may confer biological properties to nanomaterial, leading to endocytosis by cells or elimination through phagocytosis by macrophages (mononuclear phagocyte system) [146]. Moreover, it can be influenced by hydrophobicity and surface charge, committing the biodistribution of drug products [35]. The EPR effect should be studied as it can have a noticeable effect considering the high interstitial pressure presence of non-vascularized areas in the tumor microenvironment. Additionally, preclinical and clinical studies comparing conventional and nanotechnological formulations need to be performed, and not only studies comparing normal and tumor tissues to evaluate the EPR effect [146].

The aim of regulatory issues should be to prove that nanoformulations create in vivo effects, but also that they show higher efficacy and fewer adverse effects compared with conventional drugs.

## 7. Final Remarks and Future Perspectives

Polymersomes have been shown to be potential drug delivery systems due to the versatile characteristics of their block copolymers, their physical and chemical stability, and the possibility of tuning their characteristics, such as permeability and size. A wide range of block copolymers with multiple characteristics has been described, which have been improved over the years to overcome some problems associated with polymersomes, allowing targeted delivery, protection of encapsulated drugs, or selective release of therapeutic drugs. One of the most useful characteristics of the application of polymersomes as drug delivery systems is stimuli-responsive polymersomes. They are composed of block copolymers with pH, temperature, light, ROS, photo, enzyme, or redox-responsive attributes, and have demonstrated capacity to target and sequentially release therapeutic agents, being associated with high therapeutic efficiency and low toxicity. Other characteristics have been improved, namely pore formation in polymersome membranes, allowing the modulation of the permeability of nanoparticles. Properties of polymersomes, such as shape, size, hydrophilic/hydrophobic ratio, or protein fouling, can influence biological interactions and, consequently, impact the behavior and stability of polymersomes in physiological environments. These properties can be tuned through the selection of determined block copolymers and preparation methods. The size, for example, can be tuned via an extrusion step. These properties influence the EPR effect and the susceptibility to RES, efficacy, and half-life of therapeutic particles. 

Various preparation methods with high encapsulation efficiencies and the possibility of tuning polymersome characteristics, like size, morphology, or stability, have been developed. In recent years, devices and strategies have been used to improve the preparation process of polymersomes, such as microfluidics and UV crosslink strategies. Other strategies should emerge to improve methods and adapt them to the desired encapsulated therapeutics. 

Among various APIs, biotherapeutics present challenges that limit their use, such as low stability in the human body, rapid elimination, and other problems related to preparation methods. Polymersomes have been good candidates for solving these problems, protecting the therapeutic moiety, improving their efficacy, or preventing adverse effects. Polymersomes have shown potential use for many other clinical diseases in a variety of fields: diagnosis, treatment, and bioimaging. However, the most significant challenges are the development of perfect formulation methods, in terms of cost, simplicity, and effectiveness, the demonstration of the efficacy and safety of polymersomes, and the bioavailability and stability of therapeutics intended for application in the biological environment. 

Efforts should be made to improve the toxicological studies and regulatory procedures. It is important to invest in this field, as it is necessary to demonstrate the in vivo performance relative to the efficacy and toxicity of polymersomes compared with conventional drug products. In this way, it will be possible to perform better safety evaluations of drug delivery systems and facilitate translation into clinical trials. Moreover, considering that the success of clinical drug development depends on the efficacy of pre-clinical tests, the development of new strategies impacts the emergence of new drugs. So, new tests combining several types of technologies, including biotechnology, can improve our knowledge of diseases and biologic mechanisms (efficacy, metabolism, and toxicity), accelerating the development process. Patient-derived xenograft models, 3D organoids, and organ-on-a-chip are examples of new pre-clinical models that can be used in in vitro assays, testing the safety and efficacy of therapeutic products in development, necessary for translation into clinical trials. 

Despite the innovations and progress in the preparation of polymersomes, conspicuously lacking is the application of rational experimental design methods for the optimization of polymersome fabrication. While well established in the field of other types of nanocarriers, their use in polymersome fabrication is relatively rare [150]. In general, the development of polymersomes presents some barriers that are common to the different API nanocarriers. However, due to their similarities and relative advantages compared to liposomes, which are currently in clinical application, the use of polymersomes might provide an incentive for increased efforts in this field. Nonetheless, the scaling-up of manufacturing processes and the development of biodegradable and biocompatible polymers remain important areas for exploration in the future. 

## Figures and Tables

**Figure 1 materials-17-00319-f001:**
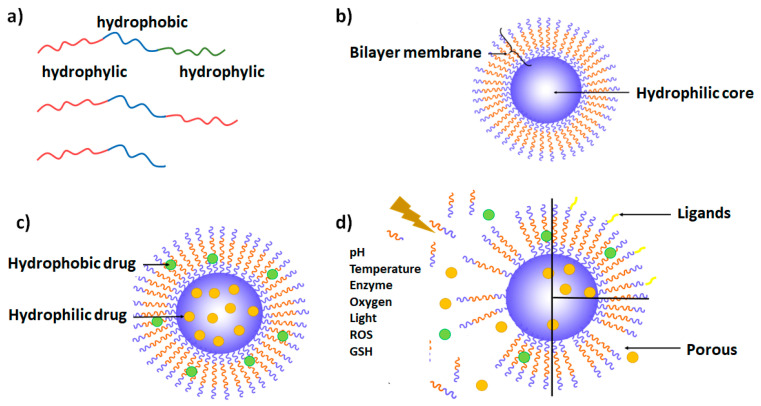
Schematic representation: (**a**) amphiphilic copolymers as polymersome building blocks; (**b**) segmented arrangement of polymersome structure; (**c**) schematic representation of a polymersome with both encapsulated hydrophilic and hydrophobic drugs; (**d**) schematic representation of stimuli-responsive polymersome, ligand-associated polymersome, and porous polymersome.

**Figure 2 materials-17-00319-f002:**
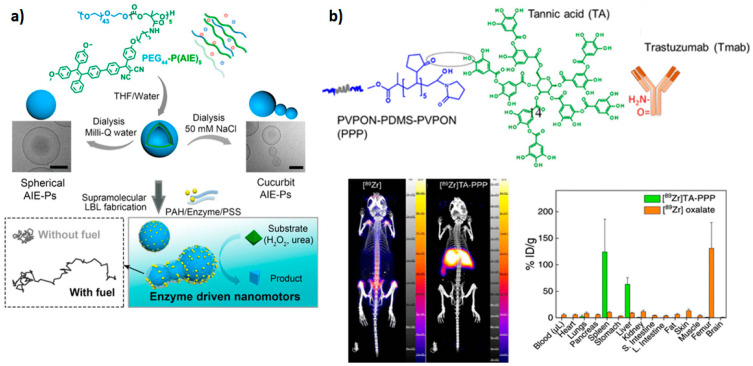
Examples of theranostic polymers: (**a**) Schematic representation of cucurbit-shaped polymersome assembly and nanomotor properties upon exposure to appropriate fuel; represented with permission from [9]. (**b**) Assembly of poly(*N*-vinylpyrrolidone)_5_-*b*-poly(dimethylsiloxane)_30_-*b*-poly(*N*-vinylpyrrolidone)_5_ polymersomes coated with tannic acid and decorated with ^89^Zr and antibody trastuzumab ([^89^Zr]TA-PPP) through hydrogen bonding and ion pairing (top). Biodistribution of free ^89^Zr and [^89^Zr]TA-PPP after 24 h. Reproduced with permission from [57].

**Figure 3 materials-17-00319-f003:**
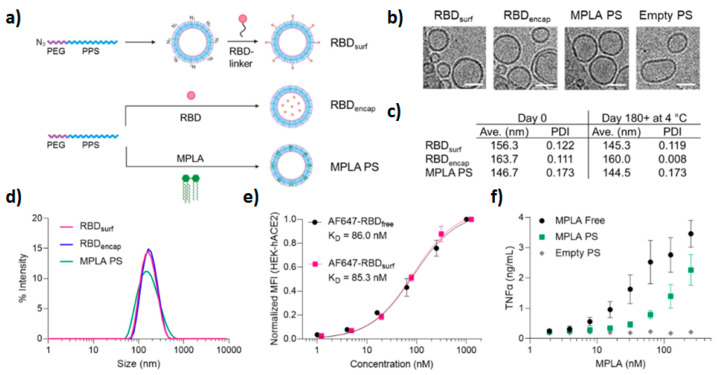
Stable polyfunctional polymersomes for delivery of antigens: (**a**) schematic of polymersome preparation and loading. Empty and RBD-loaded polymersomes were prepared using the thin film method followed by extrusion, while MPLA polymersomes were fabricated by flash nanoprecipitation followed by extrusion. (**b**) Cryoelectron microscopy of prepared polymersomes (scale 50 nm). (**c**,**d**) Respective size/size distribution curves and PDI of polymersomes. Size after six months is also indicated. (**e**) Conjugation of RBD to the polymersome surface does not influence RBD (evaluated by RBD ACE-2 receptor binding on normal human cells). (**f**) Dose-dependent secretion of TNFα when stimulated by MPLA. Reproduced with permission from [64].

**Figure 4 materials-17-00319-f004:**
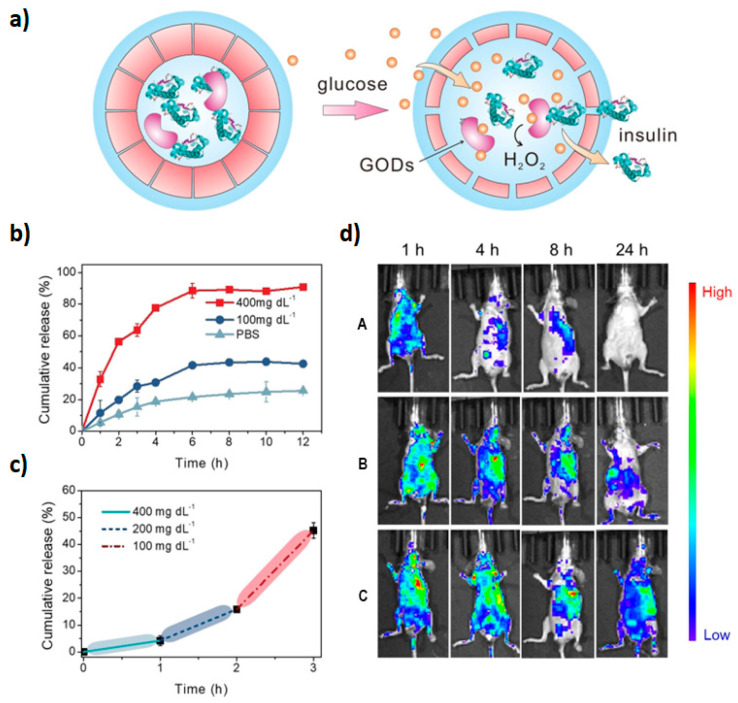
Release of insulin upon glucose stimulation: (**a**) glucose influx triggers H_2_O_2_ synthesis by glucose oxidase which, in turn, induces vesicle permeability and insulin release; (**b**) insulin release kinetics under normal and hyperglycemic conditions (100 vs. 400 mg dL^−1^); (**c**) cumulative insulin release under increasing glucose concentrations; (**d**) biodistribution of insulin within 24 h of administration in diabetic animals ((A) free insulin; (B) vesicular insulin) and normoglycemic animals ((C) vesicular insulin). Reproduced with permission from [67].

**Figure 5 materials-17-00319-f005:**
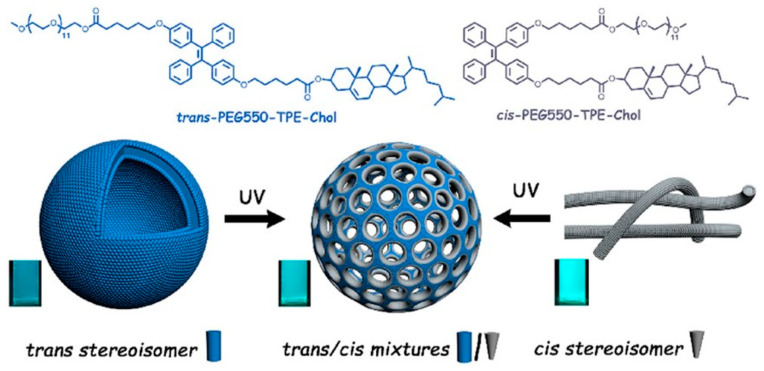
Example of a highly functionalized hydrophobic block used for the formation of light-gated polymersomes. Reproduced with permission from [48].

**Figure 6 materials-17-00319-f006:**
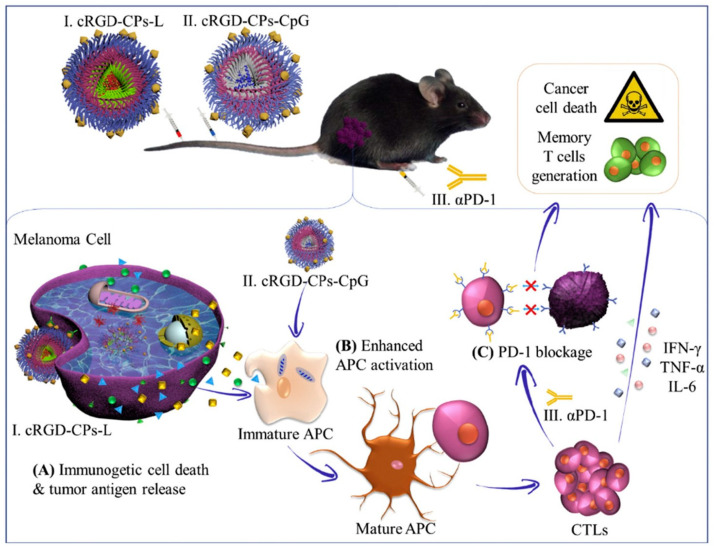
Intravenous administration of polymersomes loaded with oncolytic peptide LTX-315 (cRGD-CPs-L) and immunoadjuvant CpG (cRGD-CPs-CpG): (**A**) LTX-315 induces release of various tumor antigens, which in combination with polymersomal CpG activate maturation of antigen prese nting cells (**B**) and stimulation of T cell response; (**C**) Addition of αPD-1 antibody enhanced immunotherapeutic effect by PD-1 blockage and contributed to generation of long-term immune memory. Reproduced with permission from [25].

**Table 1 materials-17-00319-t001:** Advantages and disadvantages associated with polymersomes.

Disadvantages	Advantages
Lower biocompatibility and mimicry of cell membranes when compared to liposomes [6].	High chemical versatility due to the diversity of block copolymers [42]. Highly tunable chemical, mechanical, and stimuli-responsive (external or internal [35,36]) stability [2,6,36].
Disintegration under determined conditions (dilution or in the presence of a detergent) due to the non-covalent interactions during the self-assembly process [37].	Higher mechanical stability compared to liposomes, due to the higher molar mass of block copolymers [18], adjustable by tuning the molecular weight of the polymers [38]. Optimization of size, degradability, mechanical robustness, and encapsulation or solubilization of chemical agents [30].
Low permeability, hampering the efficient transport and release of drugs [6,36].	Possibility of tuning the permeability of polymersomes, allowing the size-selective transfer of molecules [14].
Insufficient drug loading efficiency [39].	Optimization of encapsulation efficacy; capacity to encapsulate both hydrophilic and hydrophobic drugs (due to its thicker hydrophobic membrane [6,29].
Susceptibility to the phenomenon of protein corona formation or protein fouling [40].	The use of highly hydrated, hydrophilic polymers can reduce the adsorption of large amounts of proteins [34,40]. Protection of the drugs from biodegradation, prolonging their half-life and increasing cellular uptake [11].

## Data Availability

Not applicable.

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
