# Peer review of "Polymersomes as the Next Attractive Generation of Drug Delivery Systems: Definition, Synthesis and Applications"

_materials, 2024, doi:10.3390/ma17020319_

Round 1

Reviewer 1 Report

Comments and Suggestions for Authors

The review is an excellent attempt to compile the research conducted on the polymersomes for drug delivery systems. The review is excellently written and covers a broad aspect of the recent research. Moreover, this review is very well-structured and rich in content, and I am very certain that it will attract a lot of interest from the scientific community. However, before accepting this manuscript, the authors should address some minor issues. The comments and suggestions about this work are described as follows:

1. For the title of this manuscript, it is suggested to delete “some” in “some applications”. It is obvious that the authors cannot include all the applications, and readers are not expecting to read all the applications. Therefore, there is no need to emphasize “some” applications.

2. In the Introduction, the authors used one paragraph discussing about the regulatory issues. However, from my perspective, it is out of the story. The Introduction should be a summary talking about the broad background. The regulatory issue seems a challenge that scientists should focus on in the future. Therefore, it is suggested to move this paragraph into Section 7 as a very general background.

3. In the Introductions, it is suggested to provide more information about the delivery and releasing mechanism of these polymers. It is better to briefly introduce how these polymers work and how the drug is released from these polymers.

4. In Section 5, it is suggested to provide more information and details about each method. The current descriptions are too general without some typical steps and processes.

5. The drug delivery system is a rapidly evolving field, and each passing year presents new developments that are way better than the previous ones. In such a dynamic research area, it is expected that a review must provide the latest progress in the area to justify its relevance to the field. I would suggest the authors update the review with significant reports from the last 2 years (2022 - 2023) to make it more timely and relevant.

6. In the whole manuscript, the authors only provided several figures with polymer structures. It is suggested to put more figures in the manuscript to help readers easily understand each synthetic method, recent advances, and challenges.

7. In the Conclusions and perspectives, the author can add more details focusing on describing the deficiency of the current stage of the drug delivery system and pointing out the potential or suggested direction for novel polymers in future work.

Comments on the Quality of English Language

The authors should polish their language with a more formal tongue.

Author Response

# Reviewer 1

The review is an excellent attempt to compile the research conducted on the polymersomes for drug delivery systems. The review is excellently written and covers a broad aspect of the recent research. Moreover, this review is very well-structured and rich in content, and I am very certain that it will attract a lot of interest from the scientific community. However, before accepting this manuscript, the authors should address some minor issues. The comments and suggestions about this work are described as follows:

  1. For the title of this manuscript, it is suggested to delete “some” in “some applications”. It is obvious that the authors cannot include all the applications, and readers are not expecting to read all the applications. Therefore, there is no need to emphasize “some” applications.

 Response:

Thank you for the advice. The title was corrected accordingly.

  1. In the Introduction, the authors used one paragraph discussing about the regulatory issues. However, from my perspective, it is out of the story. The Introduction should be a summary talking about the broad background. The regulatory issue seems a challenge that scientists should focus on in the future. Therefore, it is suggested to move this paragraph into Section 7 as a very general background.

 Response:

We thank the Reviewer for the observation. We have transferred the paragraph to Section 7.

  1. In the Introductions, it is suggested to provide more information about the delivery and releasing mechanism of these polymers. It is better to briefly introduce how these polymers work and how the drug is released from these polymers.

  Response:

Thank you for the advice. We have added a new paragraph to Introduction section. It now reads:

Polymersomes can be assembled from a great variety of co-polymers of different structural and physicochemical characteristics. Although certain structural parameters are required for successful polymersome formation, this variety provides almost endless possibilities for combining functional elements that enable size and shape control, targeted delivery, stimuli-sensitive drug release. With the variety of building block, a variety of fabrication techniques was developed that can complement control of polymersome characteristics by fine-tuning producing parameters and broaden the choice of loaded active therapeutic ingredients. Compartmentalized structure of polymersomes provides possibilities for encapsulation of both hydrophobic (within the hydrophobic layer) and hydrophilic (within the aqueous core or on the shell surface) molecules. Modalities that enable stabilization by polymersomes by cross-linking or are sensitive to biological of external (physical) stimuli enable control over drug release rate or spaciotemporal release.

  1. In Section 5, it is suggested to provide more information and details about each method. The current descriptions are too general without some typical steps and processes.

  Response:

Thank you for the advice. We have elaborated fabrication of polymersomes in the case of the most recent techniques, such as flash nanoprecipitation or microfluidics. Processing details are illustrated by the examples published between 2019-2023 and are labelled in cyan.

  1. The drug delivery system is a rapidly evolving field, and each passing year presents new developments that are way better than the previous ones. In such a dynamic research area, it is expected that a review must provide the latest progress in the area to justify its relevance to the field. I would suggest the authors update the review with significant reports from the last 2 years (2022 - 2023) to make it more timely and relevant.

  Response:

Thank you for the advice. We have introduced the most relevant examples and concepts developed in this decade. We believe that we have managed to present the latest and most relevant trends despite of that.

  1. In the whole manuscript, the authors only provided several figures with polymer structures. It is suggested to put more figures in the manuscript to help readers easily understand each synthetic method, recent advances, and challenges.

  Response:

We thank the reviewer for the advice. We have introduced more figures to illustrate the more recent examples.

  1. In the Conclusions and perspectives, the author can add more details focusing on describing the deficiency of the current stage of the drug delivery system and pointing out the potential or suggested direction for novel polymers in future work.

 Response:

We thank the Reviewer for the advice. In the Conclusions we have offered out opinion on future directions in the field. W have added the final paragraph which reads:

Despite the inovations and progress in preparation of polymersomes, conspicuoulsy lacking is the application of rational design methods like design-by-quality in the area of polymersome fabrication. While well established in the field of other types of nanocarriers, their use in polymersome fabrication is relatively rare. In general, development of polymersomes sufers from many impediments that are still inherent to the area of all API nanocarriers, and we have tried to summarize some of them here. However, their similarities and relative advantages over liposomes, of which some are already in clinical application, might provide incentive for increased effeorts in this area. Nonetheless, future advances are inevitably related to scale-up of favrication process and development of biodegradable e biocompatible polymers.

 The authors should polish their language with a more formal tongue.

 Response:

Thank you for the advice. We have revised the use of the language.

Reviewer 2 Report

Comments and Suggestions for Authors

This review provides a comprehensive exploration of polymersomes, encompassing their structural characteristics, preparation methods, the diverse range of employed block copolymers, and applications in the fields of biotherapeutics and cancer treatment, as well as regulatory considerations pertinent to nanodrug delivery systems. It offers some insights and concepts for the development of drug delivery systems centered on polymersomes. After addressing the following points, this manuscript will be suited for publication.

1. Some figures, like figure 1 and figure 2, in this paper are abstract. The inclusion of illustrative examples from existing literature would significantly enhance its quality. In addition, appropriate addition of some figures can make the manuscript more concrete and intuitive.

2. In light of the numerous reviews on polymersomes as drug delivery systems, clarify the unique contributions and advantages of this review compared to previously published works. Please integrate this explanation into the manuscript.

3. It appears that some references are loosely formatted, such as 'Macromolecular Bioscience, 2020, 20(6).' Ensure that references are cited accurately, and consider including more state-of-the-art references. Some reported on drug delivery and nanomedicines, such as Theranostics, 2023, 13(15), 5386-5417. doi:10.7150/thno.87854; Bioactive Materials, 2023, 26, 102-115, may be added to the revised manuscript.

4. Polymersomes find versatile applications in cancer treatment, including chemotherapy, immunotherapy, gene therapy, photothermal therapy, photodynamic therapy, and more. To enhance the comprehensiveness of this manuscript, consider addressing these applications within the text.

Comments on the Quality of English Language

Moderate editing of English language may be required.

Author Response

# Reviewer 2

This review provides a comprehensive exploration of polymersomes, encompassing their structural characteristics, preparation methods, the diverse range of employed block copolymers, and applications in the fields of biotherapeutics and cancer treatment, as well as regulatory considerations pertinent to nanodrug delivery systems. It offers some insights and concepts for the development of drug delivery systems centered on polymersomes. After addressing the following points, this manuscript will be suited for publication.

  1. Some figures, like figure 1 and figure 2, in this paper are abstract. The inclusion of illustrative examples from existing literature would significantly enhance its quality. In addition, appropriate addition of some figures can make the manuscript more concrete and intuitive.

Response:

We thank the Reviewer for the advice. We have included some illustrative examples as suggested.

  1. In light of the numerous reviews on polymersomes as drug delivery systems, clarify the unique contributions and advantages of this review compared to previously published works. Please integrate this explanation into the manuscript.

Response:

Thank you for the suggestion. Unlike most of other reviews, we have put special emphasis of development achieved in this decade. The last paragraph of Introduction now reads:

Design and application of polymersomes is a dynamic multidisciplinary field, spanning material science, technology development and biological application. We have, therefore, decide to present the development made in the field of delivery of therapeutic molecules in this decade. Furthermore, we will present some of the latest preparation methods which have been used and developed in the recent years, as well as the range of block copolymers with distinct characteristics that have been designed for polymersome preparation. Moreover, the most recent examples of their application relative to biotherapeutics delivery are represented. Finally, aspects relative to regulatory issues of nanosystems are delineated and discussed.

  1. It appears that some references are loosely formatted, such as 'Macromolecular Bioscience, 2020, 20(6).' Ensure that references are cited accurately, and consider including more state-of-the-art references. Some reported on drug delivery and nanomedicines, such as Theranostics, 2023, 13(15), 5386-5417. doi:10.7150/thno.87854; Bioactive Materials, 2023, 26, 102-115, may be added to the revised manuscript.

Response:

We thank the Reviewer for the observation and suggested articles. We have formatted the references.

  1. Polymersomes find versatile applications in cancer treatment, including chemotherapy, immunotherapy, gene therapy, photothermal therapy, photodynamic therapy, and more. To enhance the comprehensiveness of this manuscript, consider addressing these applications within the text.

Response:

We thank the reviewer for the observation. We have systematized the subsections in Section 6 to include various therapeutic fields, and have added some more recent examples.

Moderate editing of English language may be required.

Response:

Thank you for the advice. We have edited the language.

Reviewer 3 Report

Comments and Suggestions for Authors

The research topic undertaken by Authors is really interesting and worth exploring. The paper provides a lot of valuable information and constitutes a good compendium. However, Authors shoul pay more attention to the following aspects:

1) The language of the paper should be re-checked and significantly corrected.

2) The phrase "in vivo" should be written in italics.

3) The paper contains some excessive spaces (e.g. line 54) which should be removed.

4) Providing some literature references, they should be given as a range (e.g. "[23-25]" instead of "[23][24][25]") - it should be corrected in the whole paper.

5) Figures in the paper should be placed directly after the reference to them in the text (not as in the case of e.g. subsection 2.2.) - this should be improved.

6) The article contains many abbreviations thus it should be supplemented with additional subsection containing all of them and their explanations.

7) Authors did not use indentations when they start a new paragraph - it should be improved.

8) Section References should be prepared in line with the requirements of the Journal, e.g. the whole journal names should be replaced by their abbreviations. 

Comments on the Quality of English Language

The paper contains many grammar mistakes (including e.g. "the versality of copolymers make it possible..." instead of "the versality of copolymers makes it possible" - line 55 and so forth) hence it should be re-checked grammatically.

Author Response

# Reviewer 3

The research topic undertaken by Authors is really interesting and worth exploring. The paper provides a lot of valuable information and constitutes a good compendium. However, Authors shoul pay more attention to the following aspects:

1) The language of the paper should be re-checked and significantly corrected.

Response:

Thank you for the advice. We have edited the language.

2) The phrase "in vivo" should be written in italics.

Response:

We thank the Reviewer for the advice. We have formatted the text as suggested.

3) The paper contains some excessive spaces (e.g. line 54) which should be removed.

Response:

Thank you for the observation. We have edited the text as suggested.

4) Providing some literature references, they should be given as a range (e.g. "[23-25]" instead of "[23][24][25]") - it should be corrected in the whole paper.

Response:

Thank you for the observation. We have edited the references as suggested.

5) Figures in the paper should be placed directly after the reference to them in the text (not as in the case of e.g. subsection 2.2.) - this should be improved.

Response:

Thank you for the observation. We have edited the text as suggested.

6) The article contains many abbreviations thus it should be supplemented with additional subsection containing all of them and their explanations.

Response:

We thank the Reviewer for the remark. We have introduced the full name of abbreviations at the first mention in the text in concordance with the Journals instructions.

7) Authors did not use indentations when they start a new paragraph - it should be improved.

Response:

Thank you for the observation. We have formatted the text according to the Journals instructions.

8) Section References should be prepared in line with the requirements of the Journal, e.g. the whole journal names should be replaced by their abbreviations. 

Response:

Thank you for the remarque. We have formatted the references accordingly.

The paper contains many grammar mistakes (including e.g. "the versality of copolymers make it possible..." instead of "the versality of copolymers makes it possible" - line 55 and so forth) hence it should be re-checked grammatically.

Response:

Thank you. We have revised the use of English language.

Reviewer 4 Report

Comments and Suggestions for Authors

The authors present an interesting review paper, giving a overview of polymersome applications and synthesis. The work requires some mending prior to acceptance, since the placement and the use of references is not clear in many cases.

11.       Third sentence in introduction has no reference

22.       Fourth sentence in introduction has no reference

33.       Fifth sentence in introduction has no reference

44.       Page 2 line 47 no reference

55.       Page 2 line 49 first sentence no reference

66.       Page 2 line 90-84: References to respective mentioned laws and regulations missing. This is very important to point this part out, as it is of specific interest to the readers.

77.       “fcharacteristics” ????

88.       Page 6 line 220 reference missing

99.       Page 6 line 222 reference missing.

110.   Page 6 line 225 reference missing

111.   Page 6 line 228 reference missing.

112.   Page 6 line 233 reference missing.

113.   Page 6 line 235 reference missing

114.   Page 7 line 239 reference missing.

115.   Page 7 line 240 reference missing.

116.   Page 7 line 243 reference missing.

117.   Page 7 line 247 reference missing.

118.   Page 7 line 251 reference missing.

119.   Page 7 line 252 reference missing.

220.   Page 7 line 253 reference missing.

221.   Page 7 line 258 reference missing.

222.   Page 7 line 259 reference missing.

223.   Page 7 line 261 reference missing.

224.   Page 7 line 263 reference missing.

225.   Page 8 line 295 reference missing.

226.   Page 8 line 302 reference missing.

227.   Page 8 line 304 reference missing.

228.   Page 8 line 306 reference missing.

229.   Page 8 line 309 reference missing.

330.   Page 8 line 310 reference missing.

331.   Page 8 line 312 reference missing.

332.   Page 8 line 318 reference missing.

333.   Page 8 line 320 reference missing.

334.   Page 8 line 322 reference missing.

335.   Page 8 line 323 reference missing.

336.   “agents.4” Failed reference?

337.   Page 9 line 340 reference missing.

338.   Page 9 line 342 reference missing.

339.   Page 9 line 346 reference missing.

440.   Polystyrene is FDA approved. Reference should be stated.1

441.   Polystyrene is biocompatible in complexes and can be used for even drug free cancer treatment, this should be stated.2

442.   Page 9 line 346 reference missing

443.   Page 9 line 348 reference missing.

444.   Page 9 line 350 reference missing.

445.   Generally polyelectrolyte can be utilized to create polymer complexes able to encapsulate cargo without line tension, via a separate approach.3

446.   Page 9 line 354 reference missing

447.   Page 9 line 361 reference missing

448.   Page 9 line 363 reference missing.

449.   Page 9 line 364 reference missing.

550.   Page 9 line 365 reference missing.

551.   Page 9 line 368 reference missing.

552.   “ehavior.41” Different citation style.

553.   Page 10 line 393 reference missing.

554.   Page 10 line 402 reference missing.

555.   Page 10 line 404 reference missing.

556.   Page 10 line 406 reference missing.

557.   Page 10 line 412 reference missing.

558.   Page 10 line 421 reference missing.

559.   “arcinoma.10” wrong citation style.

660.   Page 10 line 430 reference missing.

661.   Page 10 line 433 reference missing.

662.   Page 11 line 447 reference missing

663.   Page 11 line 504 reference missing.

664.   Page 12 line 515 reference missing

665.   Page 13 line 530 reference missing.

666.   DMSO45

667.   Page 13 line 543 reference missing

668.   Page 13 line 557 reference missing

669.   Page 14 line 597 reference missing.

770.   Page 14 line 600 reference missing.

771.   “systemst”

772.   Page 15 line 619 reference missing

773.   Page 15 line 633 reference missing

774.   Page 16 line 671 reference missing

775.   Page 16 line 693 reference missing

776.   Page 16 line 701 reference missing

777.   Page 17 line 728 reference missing

778.   Page 17 line 742 here 2-3 references are missing since multiple studies are mentioned in this sentence

779.   Page 17 line 751 reference missing

880.   Page 18 line 776 reference missing

881.   Page 19 line 817 reference missing

882.   Page 19 line 823 reference missing

883.   Page 19 line 840 references missing.

884.   Page 19 line 857 reference missing

885.    “tranlation”???

References

(1)        FDA. FDA Drug Safety Communication: FDA recommends separating dosing of potassium-lowering drug sodium polystyrene sulfonate (Kayexalate) from all other oral drugs. Web communication. https://www.fda.gov/drugs/drug-safety-and-availability/fda-drug-safety-communication-fda-recommends-separating-dosing-potassium-lowering-drug-sodium#:~:text=Sodium polystyrene sulfonate is used,helps the body function properly.

(2)        Wu, Z.; Gao, C.; Frueh, J.; Sun, J.; He, Q. Remote-Controllable Explosive Polymer Multilayer Tubes for Rapid Cancer Cell Killing. Macromol. Rapid Commun. 2015, 36 (15), 1444–1449. https://doi.org/10.1002/marc.201500207.

(3)        Donath, E.; Sukhorukov, G. B.; Caruso, F.; Davis, S. a; Möhwald, H. Novel Hollow Polymer Shells by Colloid-Templated Assembly of Polyelectrolytes. Angew. Chem. 1998, 37 (16), 2201–2205. https://doi.org/10.1002/(SICI)1521-3773(19980904)37:16<2201::AID-ANIE2201>3.0.CO;2-E.

Comments on the Quality of English Language

minor editing neccessary

Author Response

# Reviewer 4

The authors present an interesting review paper, giving a overview of polymersome applications and synthesis. The work requires some mending prior to acceptance, since the placement and the use of references is not clear in many cases.

Response:

  1.      Third sentence in introduction has no reference
  2. Fourth sentence in introduction has no reference
  3. Fifth sentence in introduction has no reference
  4. Page 2 line 47 no reference
  5. Page 2 line 49 first sentence no reference
  6. Page 2 line 90-84: References to respective mentioned laws and regulations missing. This is very important to point this part out, as it is of specific interest to the readers.
  7. “fcharacteristics” ????
  8. Page 6 line 220 reference missing
  9. Page 6 line 222 reference missing.
  10. Page 6 line 225 reference missing
  11. Page 6 line 228 reference missing.
  12. Page 6 line 233 reference missing.
  13. Page 6 line 235 reference missing
  14. Page 7 line 239 reference missing.
  15. Page 7 line 240 reference missing.
  16. Page 7 line 243 reference missing.
  17. Page 7 line 247 reference missing.
  18. Page 7 line 251 reference missing.
  19. Page 7 line 252 reference missing.
  20. Page 7 line 253 reference missing.
  21. Page 7 line 258 reference missing.
  22. Page 7 line 259 reference missing.
  23. Page 7 line 261 reference missing.
  24. Page 7 line 263 reference missing.
  25. Page 8 line 295 reference missing.
  26. Page 8 line 302 reference missing.
  27. Page 8 line 304 reference missing.
  28. Page 8 line 306 reference missing.
  29. Page 8 line 309 reference missing.
  30. Page 8 line 310 reference missing.
  31. Page 8 line 312 reference missing.
  32. Page 8 line 318 reference missing.
  33. Page 8 line 320 reference missing.
  34. Page 8 line 322 reference missing.
  35. Page 8 line 323 reference missing.
  36. “agents.4” Failed reference?
  37. Page 9 line 340 reference missing.
  38. Page 9 line 342 reference missing.
  39. Page 9 line 346 reference missing.
  40. Polystyrene is FDA approved. Reference should be stated.1
  41. Polystyrene is biocompatible in complexes and can be used for even drug free cancer treatment, this should be stated.2
  42. Page 9 line 346 reference missing
  43. Page 9 line 348 reference missing.
  44. Page 9 line 350 reference missing.
  45. Generally polyelectrolyte can be utilized to create polymer complexes able to encapsulate cargo without line tension, via a separate approach.3
  46. Page 9 line 354 reference missing
  47. Page 9 line 361 reference missing
  48. Page 9 line 363 reference missing.
  49. Page 9 line 364 reference missing.
  50. Page 9 line 365 reference missing.
  51. Page 9 line 368 reference missing.
  52. “ehavior.41” Different citation style.
  53. Page 10 line 393 reference missing.
  54. Page 10 line 402 reference missing.
  55. Page 10 line 404 reference missing.
  56. Page 10 line 406 reference missing.
  57. Page 10 line 412 reference missing.
  58. Page 10 line 421 reference missing.
  59. “arcinoma.10” wrong citation style.
  60. Page 10 line 430 reference missing.
  61. Page 10 line 433 reference missing.
  62. Page 11 line 447 reference missing
  63. Page 11 line 504 reference missing.
  64. Page 12 line 515 reference missing
  65. Page 13 line 530 reference missing.
  66. DMSO45
  67. Page 13 line 543 reference missing
  68. Page 13 line 557 reference missing
  69. Page 14 line 597 reference missing.
  70. Page 14 line 600 reference missing.
  71. “systemst”
  72. Page 15 line 619 reference missing
  73. Page 15 line 633 reference missing
  74. Page 16 line 671 reference missing
  75. Page 16 line 693 reference missing
  76. Page 16 line 701 reference missing
  77. Page 17 line 728 reference missing
  78. Page 17 line 742 here 2-3 references are missing since multiple studies are mentioned in this sentence
  79. Page 17 line 751 reference missing
  80. Page 18 line 776 reference missing
  81. Page 19 line 817 reference missing
  82. Page 19 line 823 reference missing
  83. Page 19 line 840 references missing.
  84. Page 19 line 857 reference missing

885.“tranlation”???

References

(1)        FDA. FDA Drug Safety Communication: FDA recommends separating dosing of potassium-lowering drug sodium polystyrene sulfonate (Kayexalate) from all other oral drugs. Web communication. https://www.fda.gov/drugs/drug-safety-and-availability/fda-drug-safety-communication-fda-recommends-separating-dosing-potassium-lowering-drug-sodium#:~:text=Sodium polystyrene sulfonate is used,helps the body function properly.

(2)        Wu, Z.; Gao, C.; Frueh, J.; Sun, J.; He, Q. Remote-Controllable Explosive Polymer Multilayer Tubes for Rapid Cancer Cell Killing. Macromol. Rapid Commun. 2015, 36 (15), 1444–1449. https://doi.org/10.1002/marc.201500207.

(3)        Donath, E.; Sukhorukov, G. B.; Caruso, F.; Davis, S. a; Möhwald, H. Novel Hollow Polymer Shells by Colloid-Templated Assembly of Polyelectrolytes. Angew. Chem. 1998, 37 (16), 2201–2205. https://doi.org/10.1002/(SICI)1521-3773(19980904)37:16<2201::AID-ANIE2201>3.0.CO;2-E.

Replay:

We thank the Reviewer for the valuable advice. We have added the most of the requested references. However, during the manuscript revision process we have transferred some parts in order to achieve greater thematic coherence of Sections, and the positions of some new references are changed. Also, during the language revision phrases were reformulated and some paragraphs (examples) were removed since they were not relevant to the topic. In the cases where several phrases are related to the same example, we opted to introduce the reference in the first sentence.

Comments on the Quality of English Language

minor editing neccessary

Response:

Thank you for the advice. We have made a revision of English language.